



# The Mediterranean subsurface phytoplankton dynamics and their impact on Mediterranean bioregions

Julien Palmiéri [1,2], Jean-Claude Dutay [1], Fabrizio D'Ortenzio [3], Loïc Houpert [4], Nicolas Mayot[5], and Laurent Bopp [1]

[1]LSCE/IPSL, Laboratoire des Sciences du Climat et de l'Environnement, CEA-CNRS-UVSQ, Gif-sur-Yvette, France.
[2]Southampton University - National Oceanography Center (NOC), Waterfront Campus, European Way, Southampton SO14 3ZH, UK.
[3]LOV, Laboratoire d'Océanographie de Villefranche, CNRS, UMR 7093, Villefranche-sur-Mer, France.
[4]Scottish Association for Marine Science, Scottish Marine Institute, Oban, UK.
[5]Bigelow Laboratory for Ocean Sciences, East Boothbay, Maine, USA

**Correspondence:** Julien Palmiéri (julien.palmieri@noc.ac.uk)

**Abstract.** Ocean bioregions are generally defined using remotely-sensed sea surface chlorophyll fields, based on the assumption that surface chlorophyll is representative of euphotic layer phytoplankton biomass. Here we investigate the impact of subsurface phytoplankton dynamics on the characterisation of ocean bioregions. The Mediterranean Sea is known for its contrasting bioregimes despite its limited area, and represents an appropriate case for this study. We modelled this area using a

5   high resolution regional dynamical model, NEMO-MED12, coupled to a biogeochemical model, PISCES, and focused our analysis on the bioregions derived from lower trophic levels. Validated by satellite and *Biogeochemical-Argo float* observations, our model shows that chlorophyll phenology can be significantly different when estimated from surface concentrations or integrated over the first 300m deep layer. This was found in both low chlorophyll, oligotrophic bioregions as well as in high chlorophyll, bloom bioregions. The underlying reason for this difference is the importance of subsurface phytoplankton

10   dynamics, in particular those associated with the Deep Chlorophyll Maximum (DCM) at the base of the upper mixed layer. Subsurface phytoplankton are found to significantly impact the bloom bioregions, while in oligotrophic regions, surface and subsurface chlorophyll are of similar importance. Consequently, our results show that surface chlorophyll is not representative of total phytoplankton biomass. Analysis of the DCM finds that its dynamics are extremely homogeneous throughout the Mediterranean Sea, and that it follows the annual cycle of solar radiation. In the most oligotrophic bioregion, the total phyto-

15   plankton biomass is almost constant along the year, implying that the summertime DCM biomass increase is not due to DCM photoacclimation, nor an increase of DCM production, but instead of the "migration" − with photoacclimation − of surface phytoplankton into the DCM.





## 1 Introduction

Phenology is the study of the occurrence and characteristics of recurrent biological natural phenomena (Menzel et al., 2006). In marine science, phenology typically refers to the study of the annual cycle in phytoplankton biomass (Sverdrup, 1953; Longhurst, 2007), i.e. phytoplankton bloom starting date, the date of its maximum, bloom amplitude, bloom length, etc. These

phytoplankton dynamics or regimes are strongly constrained by − and reflect − the complex interplay between ocean dynamics (currents, temporal variability of the mixed layer depth, etc.), chemical environment (nutrients availability), and light forcing (Longhurst, 2007; D'Ortenzio et al., 2014). Therefore, phytoplankton phenology is representative of its surrounding ecosystem, and is considered as a good ecological indicator: a metric that provides objective information about ecosystem health, vigour and resilience (Platt and Sathyendranath, 2008), one that is particularly useful in regional comparisons and for characterizing

the temporal evolution of the whole pelagic ecosystem.

Phytoplankton phenology is usually defined from the temporal evolution of chlorophyll-a concentration (Chl), the latter being a good proxy of phytoplankton biomass. But because of the scarcity of *in situ* measurements, basin-scale analyses of phytoplankton phenology are almost exclusively carried out by using Chl estimated by satellite ocean color sensors ($Chl_{sat}$). Indeed, the time frequency and spatial resolution of $Chl_{sat}$ are well adapted for basin-scale analysis. In particular, $Chl_{sat}$ is

commonly used to identify ocean regions where patterns of phytoplankton phenology are similar − defined as bioregions − representing areas of the ocean with (assumed) comparable ecosystem functioning (Platt and Sathyendranath, 2008; D'Ortenzio and Ribera d'Alcalà, 2009; Racault et al., 2012; D'Ortenzio et al., 2012; Sapiano et al., 2012; Mayot et al., 2016). The concept represents useful means for marine ecosystem state assessment (Platt and Sathyendranath, 2008; Racault et al., 2014).

However, satellite-based estimates have a number of limitations, principally the remote sensing only sees the surface layer of the ocean. The first optical depth (i.e. the approximate depth seen from a space sensor) depends on water components (organic and inorganic), and it is generally limited to the first few meters (Gordon and McCluney, 1975; André, 1992). While satellite sensors can provide sufficient information for investigating phytoplankton variability in non-oligotrophic waters, in oligotrophic regions (or periods of oligotrophic conditions), the vertical distribution of the phytoplankton may not be directly

related to the observed surface signature. In particular, the presence of deep chlorophyll maxima (DCM; Cullen (1982); Berland et al. (1987); Varela et al. (1992); Cullen (2015)), ubiquitously observed in oligotrophic regimes, have been shown to coincide in the Mediterranean Sea with maximum phytoplankton biomass that is unseen in surface Chl measurements (Mignot et al., 2014). However, some indirect methods exist that permit the vertical profile of phytoplankton biomass to be estimated (Berthon and Morel, 1992; Morel and Maritorena, 2001; Uitz et al., 2006).

Hence, phytoplankton phenology − as well as ocean bioregions − estimated from $Chl_{sat}$ only reflects surface phytoplankton dynamics, and may not necessarily be representative of phytoplankton biomass dynamics, especially in oligotrophic conditions.

In this paper, we investigate the phytoplankton dynamics of the oligotrophic Mediterranean Sea (Figure 1), because it is considered a hot spot for climate change (Giorgi, 2006; The MerMex group: Durrieu de Madron et al., 2011; Diffenbaugh and



Giorgi, 2012), that endures very strong − and increasing − anthropogenic pressures Attané and Courbage (2001). Its fragile ecosystem may suffer from these strong pressures. Bioregionalization is a strong tool in this context, as it helps identifying the different ecosystems, and tracking the impact of future changes. Different recent papers focused on using annual climatologies of several physical and biological parameters, to define bioregions of the Mediterranean sea, at the surface (de la Hoz et al., 5 2018), and at different vertical layers (Reygondeau et al., 2017). A synthesis of the Mediterranean bioregions has even been performed (Ayata et al., 2018), highlighting the need of Mediterranean bioregions studies for adapted marine ecosystem management policies. But none of them analysed the sub-surface impact on chlorophyll phenology and bioregions. The second reason for studying the Mediterranean Sea is because of its particular characteristics. Despite its limited surface extent, the Mediterranean Sea is rich in phytoplankton regimes. These include productive regions such as the Liguro-Provençal sub-basin, 10 where strong spring blooms are observed, through to the ultra-oligotrophic waters of the Levantine sub-basin (D'Ortenzio and Ribera d'Alcalà, 2009; Mayot et al., 2016) that are well-known for their DCM (Berland et al., 1987; Crombet et al., 2011; Mignot et al., 2014; Lavigne et al., 2015). Moreover, the recent release of *Biogeochemical-Argo floats (BGC-Argo floats)* (Figure 2-a; Schmechtig et al. (2015)) that measure biogeochemical as well as physical variables into the Mediterranean (including Chl), presents a new avenue for insights into the occurrence and evolution of DCM in this sea (Mignot et al., 2014; Lavigne 15 et al., 2015). In particular, DCM are observed across the whole Mediterranean sea in summer (Figure 2-b), even in the northwestern region where they disappear in Winter. In general, DCM deepen eastward, from ∼40 m up to 150 m in the Levantine sub-basin, and vary on an annual cycle with a deepening in spring-summer, followed by a shallowing in autumn that tracks the depth change of an isolume (the level where the daily integrated photon flux is constant; Letelier et al. (2004); Mignot et al. (2014); Cullen (2015); Lavigne et al. (2015)). Furthermore, variations of chlorophyll concentration within the DCM are 20 attributed to a change of phytoplankton biomass, and not simply to a photoacclimation process that only changes chlorophyll (Mignot et al., 2014). Overall, DCM in the Mediterranean Sea are characterised by patterns that are already observed at global scale (Lavigne et al., 2015). The context, and these key features − the high diversity in phytoplankton dynamics in a small area; the consistent presence of a well documented DCM − serve to make the Mediterranean Sea an interesting domain to investigate the importance of subsurface dynamics for phytoplankton phenology, and its influence in the definition of the Mediterranean 25 ecosystems.

Nonetheless, *in situ* observations of the subsurface habitat and the processes there remain sparse in both space and time at this time, and preclude a comprehensive regional analysis. Recent ecological modelling studies in the Mediterranean Sea have begun to successfully represent DCM (Lazzari et al., 2012; Macías et al., 2014). Modelling then represents an attractive 30 alternative to explore questions concerning DCM.

In this study, we use a high resolution coupled dynamical-biogeochemical model (Aumont and Bopp, 2006), in a specific high resolution configuration for the Mediterranean Sea (Palmiéri, 2014), to investigate the influence of subsurface dynamics on the phenologies and bioregions of this sea. This study contributes to identifying the different ecosystems of the Mediterranean Sea − from a model point of view − in a current climate, and provides a biogeochemical model of the whole Mediterranean



Sea, available for the MerMEx community (The MerMex group: Durrieu de Madron et al., 2011) for further Mediterranean ecosystem changes study.

Here, we first present an observational validation of the model's performance, including the application of the bioregionalization procedure developed by D'Ortenzio and Ribera d'Alcalà (2009) to compare it with satellite estimates. Then, we use our simulation to investigate the characteristics of the DCM, and evaluate the impact on Mediterranean phytoplankton phenologies and bioregions.

## 2  Methods

### 2.1  The Mediterranean biogeochemical model: PISCES-MED12

The model used is the coupled dynamical biogeochemical NEMO-PISCES, in a high resolution (1/12°) regional configuration developed for the Mediterranean Sea, MED12 (Palmiéri, 2014; Richon et al., 2018c, b). PISCES is a biogeochemical model originally developed to represent the global-scale biogeochemistry, though it has previously been adapted for use in more regional studies (e.g. the Arabian Sea; (Resplandy et al., 2009, 2011)). PISCES is an intermediate complexity biogeochemical model (Aumont and Bopp, 2006), that represents 2 size class of phytoplankton, zooplankton, and particulate matter, for a total of 24 tracers. The phytoplankton growth in PISCES is limited by the availability of nutrients (nitrate, ammonium, phosphate, silicic acid, and iron). Also, uptake and release of nutrient proceeds accordingly to the Redfield ratio (O/C/N/P = 172/122/16/1; Takahashi et al. (1985)).

The NEMO-PISCES simulations here have been generated using specific Mediterranean datasets for initial and boundary conditions:

i) The model has been set-up on a 1/12° resolution grid (eddy permitting), that covers the entire Mediterranean Sea, plus a buffer zone west of the Strait of Gibraltar (see Figure 1). The dynamical model NEMO-MED12 (Beuvier et al., 2012b) was driven at the surface, using 53 years of atmospheric forcing from the ARPERA model (Herrmann and Somot, 2008), based on the ERA40-ERAInterim reanalysis for the 1958-2011 period (Uppala et al., 2005). Riverine water fluxes were prescribed from the Ludwig et al. (2009) inter-annual database. Initial temperature and salinity fields for the Mediterranean Sea were derived from the MEDATLAS II climatologies (MEDAR/MEDATLAS-Group, 2002). In the buffer zone west of the Strait of Gibralter, these variables, as well as sea surface height, were initialised by – and then strongly relaxed toward (with a timescale of 3 days) – the World Ocean Atlas (WOA) climatologies (Locarnini et al., 2006; Antonov et al., 2006). The resulting circulation fields have previously been evaluated using simulated CFC (Palmiéri et al., 2015) and $^3$He-$^3$H (Ayache et al., 2015) concentrations.

ii) The biogeochemical variables have been initialised with the Mediterranean SEADATANET climatologies (Schaap and Lowry, 2010), and from observational values (e.g. vertical cruise sections or single average values) where a spatially-resolved climatology was not available (i.e. dissolve inorganic carbon, DIC; alkalinity; iron). In the buffer zone, WOA (Garcia et al., 2006a, b) and GLODAP (Key et al., 2004) vertical profiles characteristic of the region immediately west of Gibraltar were used to initiate and relax the concentrations of biogeochemical variables. Riverine nutrient fluxes have been calculated on the





same river mouths that for the riverine fresh water in the dynamical model, and are adapted from the same dataset (Ludwig et al., 2009). No atmospheric input of nitrogen or iron (as dust) was prescribed in this simulation. An 1/8° version of this model (PISCES-MED8), also exists for climate change studies on the Mediterranean sea (Richon et al., 2018a).

The biogeochemical model was run offline following the same procedure as Palmiéri et al. (2015) and Palmiéri (2014). In those, previously simulated ocean dynamics of the NEMO-MED12 model (53 year period from 1958 to 2011 as previously mentioned) were used to drive the transport of biogeochemical tracers. After initialisation, the biogeochemical model PISCES first ran 30 years of spin-up simulation (3 loops of the 10 years − 1965-1974 − circulation fields period; the first years of the dynamical simulation − 1958-1965 −, being considered as a spin-up, are not used) in order that the biogeochemical tracers reached an equilibrium in most of the water column. After this period, the historical simulation was run covering the 1965-
2011 period available from the ARPERA atmospheric forcing. During this period, atmospheric $CO_2$ and riverine nutrient fluxes evolved following their historical records.

## 2.2 Remote sensing fields

Evaluation of modelled surface chlorophyll and calculated phenology and ecoregions makes use of satellite-based chlorophyll
estimates from Bosc et al. (2004). This product is a correction of the SeaWiFS surface chlorophyll estimates specifically developed to better represent the low surface concentrations in the Mediterranean Sea (Bosc et al., 2004). However, while decreasing considerable SeaWiFS biases, this product still over-estimates *in situ* chlorophyll measurements, in particular in the most oligotrophic parts of the Mediterranean Sea. This dataset is available for a 8 years period, from November 1997 to October 2005.

## 2.3 Biogeochemical-Argo floats

We used the recent Biogeochemical-Argo data (BGC-Argo; Schmechtig et al. (2015)) to help evaluate the modeled chlorophyll. BGC-Argo are floats transported by the current like lagrangian particles. They sample vertical profiles every 8 days, down to 1000 m depth, of temperature, salinity, nitrate, dissolved oxygen, chlorophyll, Photosynthetic Active Radiation (PAR), Coloured Dissolved Organic Carbon (CDOM), and particulate optical backscattering (bbp). In this study, only chlorophyll
is used to validate the model. The observed period with the BGC-Argo (2013-2018) being different from the one simulated by the model (1965-2012), and also because we are not interested in single profiles − but rather to validate our model with climatology on an area basis − we preferred not to perform point to point comparison for now. Instead BGC-Argo data have been gathered within 5 regions (see fig. 2), where 1) there is enough data for a reasonable data-model comparison, and 2) the regions have been chosen to be relatively close to surface chlorophyll derived clusters from D'Ortenzio and Ribera d'Alcalà
(2009). For instance: 1- Liguro-Provençal area corresponds to the *Bloom-Intermittently* cluster; 2- Algerian area corresponds to the *#3 No-Bloom* cluster; 3- Ionian area corresponds to the *Intermittently* cluster; 4- Levantine area to the *#1 No-Bloom* cluster; and 5- Tyrrhenian area that is a mix of *No-Bloom - Intermittently* clusters. All chlorophyll data have been averaged per area into monthly climatological profiles, to be easily compared to the model outputs.





### 2.4 Bioregionalization

Bioregionalization follows the method of D'Ortenzio and Ribera d'Alcalà (2009), in which bioregions are defined by applying a k-means clustering algorithm to normalized $\text{Chl}_{sat}$ annual cycles. We performed the same procedure, with some adjustments, to both modelled ($\text{Chl}_{surf}$) and satellite estimates ($\text{Chl}_{sat}$) for the period November 1997 to October 2005:

i) Satellite estimates ($\text{Chl}_{sat}$) were monthly averaged to have the same frequency as model outputs.

ii) Since available $\text{Chl}_{sat}$ are known to have artificially high chlorophyll concentration along the coast (due to high concentrations of coloured organic matter that confound the ocean colour algorithm), coastal areas are filtered from remote sensing estimates. In order to be consistent and avoid discrepancies between model and satellite estimates, filtering used model coastal area (bathymetry shallower than 160 meters) for both $\text{Chl}_{sat}$ and $\text{Chl}_{surf}$.

Finally, as we are primarily interested in the seasonal cycle of chlorophyll, we normalized all values towards specific local annual maxima (i.e. if we consider the surface chlorophyll for example, each grid cell's surface seasonal cycle is normalized to its surface annual maxima; same procedure apply when considering the DCM and the 0-300m vertically-integrated chlorophyll).

Next, the resulting normalized patterns of phenology were classified into different clusters, using a specific K-mean function
(*"kmeansruns"* from the *"fpc"* R-CRAN package) to identify different clusters in the Mediterranean Sea, each of which corresponds to a specific chlorophyll annual cycle (and is characteristic of a particular Mediterranean ecosystem). The membership of each grid cell is then mapped (regionalization) to identify regions of the Mediterranean Sea with equivalent ecosystems: the so-called bioregions.

This clusterization procedure is performed initially on surface Chl, but is additionally applied on the basis of the vertical
maximum of Chl and the vertically-integrated Chl. This may potentially result in the diagnosis of different bioregions.

The number of significant clusters is defined by analysing the stability of each cluster, and keeping the maximum number of clusters for which each resulting cluster is stable. For a fixed number of cluster, 100 "clusterization" experiments are processed. Each experiment is based on data randomly disturbed by maximum 5%. Three different disturbing method are used: Bootstrap, Noise, or Jitter (see Hennig (2007) for further details). The Jaccard similarity coefficient of a cluster (ratio of the number of
points common to both disturbed and original cluster, compare to the total number of points that belong to, at least, one of the 2 clusters) is calculated for each cluster, on each experiment. A cluster is then considered stable if its averaged Jaccard coefficient over the 100 experiments is $\geq 0.75$ (see for example table 2).

## 3    Results

### 3.1    Model Surface and subsurface chlorophyll evaluation.

The annual mean surface chlorophyll fields from both satellite-estimated and model are presented in Figure 1. The satellite estimate reveals the well-known Mediterranean characteristic of an eastward decrease of surface Chl, from the productive Alboran sub-basin ($\text{Chl}_{surf} > 1 \ \mu\text{g l}^{-1}$), to the ultra-oligotrophic Levantine sub-basin ($\text{Chl}_{surf} \sim 0.05 \ \mu\text{g l}^{-1}$). In addition,




some specific productive areas are identified: the bloom area off the Gulf of Lion in the Liguro-Provençal sub-basin associated with the formation of Western Mediterranean Deep Water (WMDW); along the flow of Atlantic Water (AW) off the Maghrebin coast; and, in the Eastern basin, the Adriatic sub-basin, the northern Aegean sub-basin, and the Rhodes Gyre (east of Crete).

The main characteristics observed in surface Chl are produced by the model. The observed eastward gradient of surface Chl,

and most of the productive areas depicted from the satellites estimates, are correctly simulated. In particular, the bloom off of the Gulf of Lion, but also the elevated surface Chl values in the Eastern basin, in the Adriatic sub-basin, and in the Rhodes Gyre (though with higher concentration than the satellite estimate). However, the AW high productivity along the Maghrebin coast is largely underestimated in the simulation. On annual average, model surface Chl globally under-estimates satellite values by a factor 2 (Figure 3). But remote sensing estimates are generally known to overestimate the surface Chl in oligotrophic waters

like the Mediterranean Sea (Bricaud et al., 2002; Claustre et al., 2002; D'Ortenzio et al., 2002; Gregg and Casey, 2004; Volpe et al., 2007; Morel and Gentili, 2009). This is also the case for our analysis, where low Chl values are especially overestimated (Bosc et al., 2004). Switching to temporal performance, the seasonal cycle is generally well simulated by the model (Figure 3), which is an imperative characteristic for conducting phenology analysis. In the Western basin, the bloom occurs in February in the model, slightly earlier than satellite estimates that present a maximum in March. Both model and satellite show minimal

Chl values in summertime (June to September), then Chl surface concentrations rise again from September, with a slower growth rate in the model. The amplitude of the seasonal cycle is less important in the Eastern basin. The model produces a broadly realistic seasonal cycle with the minima in summer, occurring however slightly later, and for a longer period, than in satellite estimates. There is also a slower and later increase toward the winter maximum values, indicating some difficulty for the model in simulating autumn phytoplankton growth. As noted previously, in this study we will pay attention to the influence

of subsurface Chl. Thus, we now evaluate our model results with *in situ* observations on vertical profiles. In this purpose, we use chlorophyll profiles of the recent BGC-Argo floats frequently release in the Mediterranean Sea since 2013 (Schmechtig et al., 2015) . BGC-Argo and model chlorophyll profiles are compared figure 4 in the 5 areas defined figure 2. It shows that the model succeeds in capturing the overall chlorophyll dynamics. In particular, it confirms what has just been deduced from $Chl_{surf}$ comparison with remote sensing estimates: the general behaviours are present, but the model underestimates surface

chlorophyll by ∼70% compare to BGC-Argo observations (annual average of the 5 areas, see table 1). Below, the subsurface chlorophyll seasonal dynamics are also well represented, with a summer deepening (on average from 56 m in winter to 140 m in summer) and strengthening of the DCM (in average from 0.17 in winter to 0.39 $\mu$g-Chl $l^{-1}$ in summer), modeled in all 5 areas. Yet, *BGC-Argo* presents $Chl_{max}$ maxima in spring within 2 of the 5 areas: in the Gulf of Lion, and Algerian area (what is not present in the model, as the surface chlorophyll is underestimated − see the discussion, section 4.3, and the appendix

A). The regional features are also modeled. PISCES-MED12 manifests an eastward deepening (from 103 to 186 m depth in summer) and weakening of the DCM (in summer, from 0.58 in the Gulf of Lion to 0.22 $\mu$g-Chl $l^{-1}$ in the Levantine sub-basin), what is all − qualitatively − in good agreement with the observations. However this comparison also highlights that the model underestimates the Chl concentration, not only at the surface, but also at depth by ∼60%. It suggests that the model underestimates the chlorophyll in general (see the appendix A, for further evaluation and discussion on this point). Furthermore

DCM depth is always deeper than observed by approximately ∼30 to ∼50 m.





Finally, despite some shortcomings compared to observations, the model generally produces a global Chl distribution with realistic temporal and spatial structures. Notwithstanding the mismatches in absolute chlorophyll identified above, the overall performance of the model is sufficient to permit an investigation of the general patterns of chlorophyll phenology in the Mediterranean Sea (see discussion, section 4.2).

## 3.2 Surface chlorophyll phenologies and bioregions

Applying the clusterization procedure to both $\text{Chl}_{sat}$ and model $\text{Chl}_{surf}$ results in different total cluster numbers (see Table 2). Model Chl results in 4 clusters, while satellite Chl results in 5. For parsimony, we then use the common 4 clusters in the rest of the study for both model $\text{Chl}_{surf}$ and $\text{Chl}_{sat}$ in order to enable further comparisons.

The 4 clusters identified from the remote sensing estimates (based on a monthly dataset; see Figure 5-a) have strong similar-
ities with the original analysis by D'Ortenzio and Ribera d'Alcalà (2009) (where 8-day data frequency was used, and 7 clusters were generated; see their Figure 4). These are comparable in both their spatial distributions, and their respective chlorophyll annual cycles. Differences appear due to the preliminary treatment we performed on the dataset (see section 2.4 above) and also to the number of clusters we imposed for consistency with model $\text{Chl}_{surf}$. For instance, the 2 coastal bioregions of D'Ortenzio and Ribera d'Alcalà (2009) are logically no longer present, given the coastal filtering that we applied. Also, limiting the num-
ber of cluster to 4 resulted in the merging of the *Bloom* and *Intermittently* regimes of D'Ortenzio and Ribera d'Alcalà (2009) into a single *Bloom-Intermittently* bioregion (in red in Figure 5-a). The seasonal cycles of the remaining 4 clusters are very similar to those from D'Ortenzio and Ribera d'Alcalà (2009), with minimum values occurring in every cluster in summer (JJA). The *Bloom-Intermittently* regime is characterised by high and pronounced maximum values in March, while the 3 other clusters produce smoother maxima in winter (DJF). These latter regions can be considered − following D'Ortenzio and Rib-
era d'Alcalà (2009) analysis − as 3 *No-Bloom* bioregions. The normalized chlorophyll used here, provide no information about the absolute chlorophyll concentration in each cluster, but they indicate the relative amplitude of chlorophyll seasonal cycles. This amplitude is greatest for the *Bloom-Intermittently* bioregion, which shows a large spring increase in chlorophyll while other *No-Bloom* bioregions present much more moderate seasonal amplitudes.

The *Bloom-Intermittently* bioregion covers most of the Liguro-provençal sub-basin, as well as the northwestern part of the
Tyrrhenian sub-basin, east of the Bonifacio Strait. The 3 *No-Bloom* regimes are located at similar places as in D'Ortenzio and Ribera d'Alcalà (2009): one is in the Adriatic and Aegean sub-basins, and around the Alboran gyres (in yellow); a second one in the southern part of the Algerian-Provençal sub-basin that extents eastward, along the Sicily strait and in the southern Tyrrhenian up to the south-Western part of the Ionian sub-basin (in green); and the third one extends into the Ionian, Levantine and Tyrrhenian sub-basins, and along the south-eastern coast of Spain (in blue).

When applied to the model outputs, the clusterization procedure produces analogous results to those of the satellite estimates (see Figure 5-b). It generates the main structures − one *Bloom-Intermittently* bioregion (red), and 3 *No-Bloom* − with similar annual cycles that have realistic winter-spring maxima and summer minima in chlorophyll. However, despite these robust similarities, some differences obviously exit. For instance, the amplitudes of the seasonal cycles tend to be more elevated than those deduced from $\text{Chl}_{sat}$, due to overly oligotrophic conditions in summer in the model (Palmiéri, 2014). Simulated autumn



$\text{Chl}_{surf}$ increases later, due to a nutricline in the model that is too weak, especially in the western basin (see the appendix A). Model blooms also occur slightly earlier (February-March) than those observed (April).

The clusters' spatial structures also reveal some differences. The modelled *Bloom-Intermittently* bioregion is satisfyingly located in the Gulf of Lions, but it extends to the east, taking in the Northern Ionian, the Southern Adriatic, and the Levantine

sub-basins, while it is excluded from the Tyrrhenian sub-basin. The *No-Bloom* bioregion observed off the Maghrebin coasts in $\text{Chl}_{sat}$ based results (see the green cluster in Figure 5-a), has its extension reduced in the model to the Alboran sub-basin and the western part of the Algero-Provençal sub-basin. In the model the *No-Bloom* cluster observed in the Adriatic and Aegean sub-basins (yellow) is extended along coastal regions of almost all of the eastern basin, the Eastern Tyrrhenian sub-basin, and the western part of the Alboran sub-basin.

In summary, despite some unavoidable differences, when applied to the model surface chlorophyll, the clusterization procedure generates results that are consistent with those from satellite estimates. The model produces coherent bioregions, with realistic chlorophyll annual cycles/phenology from surface chlorophyll fields, even if their respective geographical location may differ from those of satellite estimates. Bearing in mind the limitations associated with these model-observation discrepancies, using the model to study the phenology and bioregions of the Mediterranean Sea remains plausible.

**3.3  Phytoplankton dynamics in the whole epipelagic layer**

Surface chlorophyll is commonly used as a proxy of phytoplankton biomass, a key facet of interest for biogeochemists. However, to our knowledge, very few studies link the surface annual Chl signal to the integrated Chl for the whole epipelagic layer − and these are still early stage studies (Uitz et al., 2006; Alvain et al., 2008; Uitz et al., 2012). In part, as implied above, this stems from the limited availability of appropriate observational data. Here, we take advantage of complete model fields to

address the issue of the relationship between surface and deep chlorophyll. Applying the clusterization approach to chlorophyll integrated across the $0 - 300\,\text{m}$ deep layer ($\text{Chl}_{tot}$; a layer wide enough to include the euphotic layer). produces bioregions and corresponding annual cycles (Figure 6) that are very different from those obtained using $\text{Chl}_{surf}$. The procedure generates the same number of stable clusters (Table 3), but there are a number of significant differences. In particular, the temporal structure of all clusters is significantly modified, with annual cycle amplitudes largely reduced, and the appearance of 2 local maxima.

The first one occurs in February-March, as with $\text{Chl}_{surf}$, but an additional maxima now manifests in summer. Such large modification can only be attributed to subsurface chlorophyll. Furthermore, unlike surface chlorophyll, for which annual minima happen in summer, total chlorophyll minima now occur in November-December across all bioregions. The annual cycles of all $\text{Chl}_{tot}$ regimes present a progression from bioregions where the seasonal amplitude is relatively high, and where the late winter-early spring signal is dominant (in red), through to bioregions where the amplitude is very low, and where the summer

signal is slightly dominant (in blue).

Furthermore, the locations of the $\text{Chl}_{tot}$ bioregions differ from those of $\text{Chl}_{surf}$. These locations also present the progression highlighted in $\text{Chl}_{tot}$ phenology, from the red to the blue cluster. Interestingly, the locations of these only 2 extreme bioregions are very similar to the 2 $\text{Chl}_{surf}$ extremes: the red *Bloom-Intermittently* and the yellow *No-Bloom* (Figure 5-b). But the red $\text{Chl}_{tot}$ bioregion's area is less expended than from the surface, and the blue $\text{Chl}_{tot}$ does not include the Aegean sub-basin.





Hence both positions and phenologies are different from what is observed at the surface. That result was foreseen for the most oligotrophic part of the Mediterranean sea, but the model shows that $Chl_{tot}$ phenology is also different from $Chl_{surf}$ one in the *Bloom-Intermittently* bioregion.

### 3.4 Phytoplankton dynamics in the deep chlorophyll maximum

The differences between $Chl_{surf}$ and $Chl_{tot}$ phenologies are necessarily attributable to subsurface chlorophyll. The evaluation of PISCES-MED12 simulation demonstrated that the model represents a marked DCM signal (Figure 4 ). We now describe the large scale spatial and temporal distribution of the DCM to examine how it influences the epipelagic ($0 - 300$ m) integrated Chl distribution ($Chl_{tot}$).

A seasonal analysis of the simulated chlorophyll distribution (Figure 7) reveals that maximum concentrations are generally not at the surface, with the exception of the winter in some restricted regions in the Alboran, Adriatic, and Levantine sub-basins, and in the Gulf of Lion. In addition, seasonal chlorophyll maxima no longer occur in winter/spring, but instead in summer when the Mediterranean Sea is at its most oligotrophic. The depth of the DCM deepens eastward, from below 50 meters in the western part of the Mediterranean Sea, down to ~180 meter depth in the eastern part. Simulated deep chlorophyll

maximum depth also varies seasonally, from a minimum in winter-spring to a maximum in summer-autumn, consistent with DCM observations (Crombet et al., 2011; Mignot et al., 2014; Lavigne et al., 2015). Chlorophyll concentrations in the modelled DCM, as well as its seasonal depth progression, are generally realistic, with the exception of the DCM depth, that is deeper − by about 30 to 50 m − in the model.

Applying the bioregionalization procedure to the DCM signal provides new insights for the phenology of the Mediterranean

Sea (Figure 8). Model $Chl_{max}$ exhibits less complexity than $Chl_{surf}$, producing only two stable clusters (see table 4). One (in green) covers the western basin plus the areas of the eastern basin where $Chl_{surf}$ has its red *Bloom-Intermittently* bioregion (see the red cluster in Figure 5-b); and the second one (yellow) extends across most parts of the eastern basin. The annual cycles differ completely from those of the corresponding surface bioregions. The dynamics of $Chl_{max}$ follow the seasonal solar radiation cycle, with a minimum in winter (January or February) and a maximum in summer (June or July), suggesting

that phytoplankton growth in DCM is mainly controlled by solar energy rather than nutrient availability. Differences between the "Western" and "Eastern" clusters reside mainly in the amplitude of the $Chl_{max}$ annual cycle, which is less important in the "Eastern" cluster (yellow), due to its more oligotrophic conditions. The amplitude of the western cluster is more important because this region benefits from higher nutrient supply, due to mixing with rich intermediate and deep water, and to the inflow of nutrient-rich Atlantic Water at the surface. These higher nutrient concentrations enable a shallower DCM and a higher receipt

of solar energy by the phytoplankton, both of which result in higher chlorophyll concentrations.



## 4 Discussion

### 4.1 Bioregions and Mixed Layer Depth

The *Bloom-Intermittently* cluster made from $Chl_{Surf}$ (Figure 5) is the most recognizable due to its specific phenology and locations. Modelled phenology is correct for this cluster, but its locations differ from that derived from remotely sensed $Chl_{Surf}$.

Spring-blooms are induced by the high amount of nutrients introduced to the euphotic layer by mixing, from deep and intermediate waters. Furthermore, the interplay between the Mixed Layer Depth (MLD) and euphotic depth, can induce a temporal lag between the establishment of conditions favouring blooms, and bloom occurrence (Sverdrup, 1953; Dutkiewicz et al., 2001; Lavigne et al., 2013; D'Ortenzio et al., 2014). As MLD is a key factor for spring bloom occurrence, the modelled MLD may explain the differences between the satellite and the model *Bloom-Intermittently* cluster.

Indeed, while the circulation model NEMO-MED12 exhibits realistic deep water formation in the Gulf of Lion (in the Liguro-Provençal sub-basin; Beuvier et al. (2012a)), some defects are known in this simulation in the Eastern basin: 1. the formation of the Levantine Intermediate Water (LIW) is too intense; and 2. NEMO-MED12 has excessive ventilation of intermediate waters in the Ionian sub-basin instead of the Adriatic deep water formation (Ayache et al., 2015).

    Applying a regionalization procedure (with 4 clusters like with chlorophyll) to both a Mediterranean climatology (Houpert

et al., 2015) and NEMO-MED12 MLD (see Figure 9), results in a cluster (in red) located exactly in the same regions that the *Bloom-Intermittently* chlorophyll cluster. This red MLD cluster is associated with the Mediterranean blooms in both model and satellite estimates, and presents exactly the same differences between the model and the observations. It confirms that the differences seen between the model and the satellite *Bloom-Intermittently* bioregions are caused by differences in the circulation model dynamics.

Differences within the other $Chl_{surf}$ clusters between satellite and model results (Figure 5) are more difficult to investigate. At first glance, the physical dynamics do not appear to be the main determinant factor. MLD clusters in the model and in the climatology (except the red one discussed immediately above) are very similar, and hence cannot explain the differences in the *No-Bloom* clusters. Thus, these differences may be the effect of other factors highlighted in PISCES-MED12 results like a too weak nutricline in western basin, or too low phosphate concentrations over the first 100 m depth of the Mediterranean sea (see

Appendix A and Palmiéri (2014)).

### 4.2 Modelled chlorophyll maximum and total chlorophyll phenologies

In order to evaluate in more detail the simulated Chl phenology, we compare it to the novel information gained from the recently released *BGC-Argo* floats (Schmechtig et al., 2015)). We averaged *BGC-Argo* Chl profiles to get a monthly averaged annual

cycle in the 5 areas defined in figure 2, and normalized them by their respective annual maximum. In each area, we selected specific layer from the vertical profile: the surface layer ($Chl_{surf}$), the Chl maximum ($Chl_{max}$ − which is not necessarily deep − and its depth), and the vertical sum of Chl ($Chl_{tot}$; $0 − 300$ m, as done in the model), from which we derived a normalized annual cycle (Figure 11). This procedure has been done to both BGC-Argo (left column - Figure 11), and model outputs (right





column - Figure 11). First, across all selected areas it produces a surface chlorophyll signal (in green) that is comparable with that estimated from satellite (Figure 5). It gives confidence to our procedure for treating the information from the Argo floats. The difference with simulated $Chl_{surf}$ confirms the model difficulties to reproduce the autumnal surface chlorophyll increase (Figure 3), which appears too late, and too weak, because of a generally low $PO_4$ surface concentration (see the appendix A).

The vertical Chl maxima obtained from the Argo float observations (in red) is globally similar to that produced by the model. Its phenology presents more variability, but the general $Chl_{max}$ dynamics confirm a maximum in summer and minimum in winter, with the exception of the Algerian sb, where $Chl_{max}$ is quite flat all over the year at the exception of a low value in December (but this might be due to the small number of observations in this sub-basin, or reflects sub-basin peculiarities such as its intense meso-scale eddy activity). The second exception is in the Gulf of Lions, where the spring bloom significantly

dominates the $Chl_{max}$ phenology with an amplitude of 0.8 in the Liguro-Provençal sub-basin (what is not seen in the model, as − even if the bloom is modeled, and the bloom cluster appears − its amplitude is largely underestimated), that compares to $0.4-0.5$ in other areas. These apparent $Chl_{max}$ discrepancies could possibly be attributed to the non-homogeneity in *BGC-Argo* data, itself caused by the relatively short time records to date of the Lagrangian floats, and/or by the non homogeneity of the lagrangian floats sampling method itself. However, the elevated $Chl_{max}$ during the bloom period in the Liguro-Provençal

sub-basin is most likely a real signal.

    Despite the *BGC-Argo* float data being at an early stage, it still provides some interesting information on phenology that is reproduced by our simulation. For instance, $Chl_{tot}$ phenology is different everywhere from that of $Chl_{surf}$. Also, the sub-basins have a range of different seasonal regimes, with a progression from a surface-dominated regime with high phenology amplitude (0.7) in the Liguro-Provençal sub-basin through slightly surface-dominated situations in the Algerian, Tyrrhenian and Ionian

sub-basins (with interesting differences concerning winter to summer local maxima), to a more equilibrated surface/subsurface regime with low phenology amplitude (0.2) in the Levantine sub-basin (i.e: in the most oligotrophic area). As already noted, all of these characteristics derived from the *BGC-Argo* profiles are in agreement with our modelling results. Only differences compare to model $Chl_{tot}$ are likely to be related to the underestimated surface chlorophyll in the western basin. The shape is good in the Liguro-provençal sub-basin, but the dominance of the bloom is underestimated. As well, the maximum $Chl_{tot}$ is

in summer in the Algerian and Tyrrhenian sub-basins, with a too flat $Chl_{tot}$ phenology (amplitude of ∼0.2 compare to ∼0.4), when surface signal should be the slightly dominating one.

## 4.3   Underestimated chlorophyll and DCM depth.

This study has shown that the model underestimates the chlorophyll at both the surface, and the DCM, and that it simulates a DCM that is too deep (see discussion of the later issue in the appendix A). How can this affect the $Chl_{tot}$ phenologies and

resulting bioregions?

    First, we use normalized chlorophyll, as done by D'Ortenzio and Ribera d'Alcalà (2009) and Mayot et al. (2016), in part because we are interested in the chlorophyll variation along the year, but also to avoid chlorophyll bias to impact the results. Regarding $Chl_{tot}$, it won't have any consequences, as far as the bias is the same in the whole water column, that $Chl_{surf}$ and $Chl_{max}$ are given the same weight in their contribution to $Chl_{tot}$, than observed. That is the case in the Eastern basin: There,




$Chl_{surf}/Chl_{max}$ is 0.3 on average in BGC-Argo data, and 0.28 in the model (table 1). In the Western basin, we know it is not the case. $Chl_{surf}$ contribution is underestimated, with an observed $Chl_{surf}/Chl_{max}$ of 0.45, and of 0.28 in the model (table 1). It explains why the modelled $Chl_{tot}$'s surface signal is not dominant in the Algerian and Tyrrhenian sub-basin, and not enough dominant in the Liguro-Provençal. That may have an impact on the bioregion definition of the Western basin.

The too deep DCM is less problematic for $Chl_{tot}$. As we are interested of the monthly variation of the vertical chlorophyll sum, as far as the $Chl_{surf}$ / $Chl_{max}$ balance is respected (as discussed above), the depth of $Chl_{max}$ will not have any impact on $Chl_{tot}$ phenology, or the induced bioregions. The definition of $Chl_{tot}$ is independent of the chlorophyll depth. It is still important to notice and investigate as it highlights a default in the simulation (here the nutricline is too smooth, see the appendix A), but a better $CHL_{max}$ depth would not improve $Chl_{tot}$.

As the number of *BGC-Argo* floats is still regularly increasing in the Mediterranean Sea, the spatial and temporal coverage of the float database will − probably within a few years − be large enough to derive an appropriate 3D climatology of the depth distribution of Mediterranean chlorophyll. This potentially represents an important alternative for investigating ecosystems of the Mediterranean Sea compared to both *in situ* observations and satellite-estimated surface chlorophyll.

### 4.4 Phytoplankton dynamics in the oligotrophic bioregion.

The preceding sections highlighted that the bioregions and phenology defined from $Chl_{surf}$ and $Chl_{tot}$ differ, and that these differences are strongly impacted by the behaviour of the DCM. The DCM results from the interplay between light and nutrient availability. It is situated at the top of the nutricline, where the balance of downward photon flux and upward nutrients flux is optimal (Mignot et al., 2014; Cullen, 2015; Lavigne et al., 2015). The DCM experiences a seasonal vertical displacement, following the isolumes (Letelier et al., 2004; Mignot et al., 2014; Cullen, 2015; Lavigne et al., 2015) that deepen in summer

with solar radiation annual cycle, and also with the summertime disappearance of the surface chlorophyll-induced shadow effect. Mignot et al. (2014) suggest that the DCM is a result of photoacclimation processes (an increase of the phytoplankton Chl/C ratio when available light decreases) that result in a higher Chl/C ratio in the DCM than in surface waters (this has been confirmed by a modelling approach Ayata et al. (2013)). However, they also revealed that, inside the DCM, the seasonal $Chl_{max}$ concentration variations do not result from a further photoacclimation process (as DCM is supposed to follow an isolume) but

rather result from changes in the phytoplankton biomass inside the DCM.

This photoacclimation process occurs in our simulation since the biogeochemical model PISCES includes the photo-adaptative model of Geider et al. (1997). It is illustrated when analyzing the phytoplankton biomass bioregionalisation (Figure 10), which results in 4 clusters (Table 3) that are extremely similar to those from $Chl_{tot}$ (Figure 6). The only difference is due to phytoplankton photoacclimation. In particular, in the most oligotrophic region (blue cluster), the phytoplankton biomass is

almost constant from February to August, while $Chl_{tot}$ reveals a Chl increase on the same period. Hence, this difference does not result from any global increase of the phytoplankton biomass in this bioregion, but rather from a vertical "migration" − not in sense of individual movement, but rather understood as the "movement" of favourable growth conditions − and gathering of the phytoplankton − with its related photoacclimation − into the DCM.



### 4.5 Surface versus total chlorophyll bioregionalization

Unsurprisingly, surface and total chlorophyll phenologies and bioregions are different. In particular, phenologies highlight the fact that both surface − maximum peak in late winter to early spring − and subsurface chlorophyll dynamics − maximum peak in summer − are discernible and have a significant impact on $Chl_{tot}$ phenology across all bioregions. Surface Chl is characterized by a minimum in summertime and a maximum in late winter-early spring that are driven by physical dynamics and the related winter vertical mixing that brings nutrients to the surface. In contrast, the subsurface signal is mainly controlled by the annual cycle of solar radiation that generates maximum Chl concentrations in the summer season. The integrated signal ($Chl_{tot}$) reveals characteristics that results mainly in a combination of these two modes of variability, with characteristics of both. This result was expected for the most oligotrophic part of the Mediterranean Sea, but the model shows that $Chl_{tot}$ phenology is also different from the $Chl_{surf}$ one in the *Bloom-Intermittently* bioregion.

Most of the ocean biogeographical studies are based on $Chl_{surf}$, with the hypothesis that the surface chlorophyll is a good proxy of the total phytoplankton biomass (D'Ortenzio and Ribera d'Alcalà, 2009; Racault et al., 2012; D'Ortenzio et al., 2012; Sapiano et al., 2012) the latter being of greatest interest for biogeochemistry. To our knowledge, only one study (Platt et al., 2009, 2010) compared surface chlorophyll phenology to phytoplankton biomass, with both being estimated by remote sensing (the latter calculated using an empirical model estimating phytoplankton biomass from $Chl_{sat}$). It highlighted that new features, unseen by the $Chl_{sat}$ appeared on the phytoplankton phenology, in particular in low latitudes (Platt et al., 2009). However, the comparison has not to date been extended to biogeography comparisons.

When using $Chl_{tot}$, the bioregions definition changes from bioregions defined on surface characteristics, to bioregions defined on a surface-to-subsurface Chl balance. The main reason for these differences being that subsurface productivity is shown to be important, principally in oligotrophic regions (as expected) but also in the bloom areas of the Mediterranean sea. Bioregions based on $Chl_{tot}$ generate a more continuous bioregionalization, with an intermittently bioregion (yellow) now marking the transition between the bloom bioregion (red; strong surface signal) and the *No-Bloom* bioregions (green and blue, the most oligotrophic; equivalent amplitude in both surface and subsurface Chl signal). Our results show that in the Mediterranean Sea − a domain with a wide range of phytoplankton dynamics − the total chlorophyll is uniformly much closer to phytoplankton biomass (Figure 10) than it is to surface chlorophyll, with significant consequences for the results and induced interpretations.

Interestingly, the 2 extreme bioregions (*Bloom-Intermittently* and the most *oligotrophic*) are the only ones that present reasonably close clusters location, within both $Chl_{surf}$ and $Chl_{tot}$. This could have different reasons. 1- The surface signal is sufficiently dominant to keep being the main signal on $Chl_{tot}$. This might be plausible regarding the *Bloom-Intermittently* bioregion, but not for the most *oligotrophic* one. Or 2- in these bioregions, surface and subsurface evolve together, and are more strongly correlated, getting a coherent and geographically identifiable cluster from either surface or vertically integrated chlorophyll signal.

Thus, our results show that the surface chlorophyll alone is not a good proxy of total phytoplankton biomass. This does not, of course, mean that the surface chlorophyll is incorrect, but merely that it is incomplete: subsurface features that can completely change the interpretation of phytoplankton dynamics, and hence of the epipelagic ecosystem, are not taken into account (we



know since Reygondeau et al. (2017) that epipelagic, mesopelagic and bathypelagic bioregions, and hence ecosystems, are different). This is obvious and already known, but marine ecosystem studies using remote sensing surface chlorophyll estimates already interpret their results as if they were representative of the whole epipelagic ecosystem. Phenologies and bioregions defined from surface chlorophyll are characteristic of the surface phytoplankton dynamics only, and cannot be said to represent
the whole epipelagic ecosystem.

Nonetheless, recent remote sensing products may still help with marine ecosystem analysis, for instance the community structure products of Alvain et al. (2006) and Uitz et al. (2006). Another solution could be using remote sensing derived primary production products (see for example for the Mediterranean sea Bricaud et al. (2002); Bosc et al. (2004); Uitz et al. (2012). But one must be very prudent using these primary production products, as it has been shown that depending on the
algorithm, the resulting primary production can have up to a factor 2 of difference (Carr et al., 2006; Saba et al., 2010).

## 5   Conclusions

Phenologies and bioregions of the Mediterranean Sea have been analysed using the biogeochemical model PISCES in a high-resolution, Mediterranean configuration, MED12. The model simulates realistic surface chlorophyll, which reproduces the
main characteristics of satellite estimates, as well as deep chlorophyll maxima with the correct geographical and temporal patterns.

Bioregions defined from model surface chlorophyll are coherent, with patterns of phenology that are close to satellite estimates, and include the main observed characteristics. However, the geographical positions of bioregions are quite different.
In particular, the *Bloom-Intermittently* cluster overlaps that derived from satellite estimates, but has differences in its spatial extent − especially in the Eastern basin − caused by formation of deep and intermediate water that is too strong in the North Ionian-Adriatic and in the Levantine sub-basins.

Taking into account the whole epipelagic layer causes changes in both phenology and bioregions. Integrated Chl bioregions
are primarily defined by a surface to subsurface signal balance, and present a progression from the bloom bioregion to the most oligotrophic one.

The results illustrate the importance of subsurface dynamics − and of the Deep Chlorophyll Maximum in particular − in the Mediterranean Sea. A bioregionalisation of the DCM revealed a very homogeneous annual cycle of its chlorophyll content, with seasonal progression that evolves with the annual solar radiation cycle. Furthermore, in *oligotrophic* areas, stable phytoplankton
biomass implies that the summer Chl increase in the DCM results from a phytoplankton "migration" from the surface layer to the DCM. This highlights the misunderstanding that comes from surface Chl analysis. The phytoplankton biomass does not decrease drastically in summer in *oligotrophic* bioregions because of nutrient limitation. Instead, it mostly goes deep to the DCM, where growth conditions are adequate with higher nutrients concentration and sufficient light availability.



Finally, we show that phenology and bioregions based on $Chl_{surf}$ are very different from those based on integrated phytoplankton biomass. It appears that surface chlorophyll alone is insufficient to describe epipelagic ecosystem functioning, and hence cannot be considered as a good proxy of the whole phytoplankton biomass (at least in the Mediterranean). So phenology and bioregional studies that are based on remotely sensed surface chlorophyll estimates are only representative of surface plankton dynamics and not the whole epipelagic ecosystem.

## 6   Acknowledgements

This study was conducted as a part of the WP1 MerMEx/MISTRALS project. This paper is a contribution to the international SOLAS, IMBER and LOICZ projects. The authors would like to thank: The MerMEx project, the Remotely Sensed Biogeochemical Cycles in the Ocean (remOcean) project, funded by the European Research Council (grant agreement 246777), and the French BGC-Argo project funded by CNES-TOSCA, and to the French "Equipement d'avenir" NAOS project (ANR J11R107-F). We also thank Andrew Yool for his help in improving the English of this paper.





**Table 1.** Seasonal average of surface, and vertical maximum of chlorophyll concentration (in $\mu$g l$^{-1}$) and of the depth of this maximum (in meter), from the vertical profiles Fig. 4), calculated for the 5 areas defined Fig. 2, for both *BGC-Argo* in situ observations, and the PISCES model. Western region values take in account the Golf of Lions, the Algerian, and the Tyrrhenian regions, and Eastern region values the Ionian and the Levantine regions.

| | | Spring | Summer | Autumn | Winter | Annual |
|---|---|---|---|---|---|---|
| Golf of Lions | Chl$_{surf}$ ($\mu$g l$^{-1}$) | 1.30 (0.20) | 0.09 (0.04) | 0.20 (0.06) | 0.58 (0.27) | 0.54 (0.14) |
| | Chl$_{max}$ ($\mu$g l$^{-1}$) | 1.71 (0.45) | 1.18 (0.58) | 0.63 (0.40) | 0.59 (0.27) | 1.03 (0.43) |
| | max depth (m) | 29 (56) | 49 (98) | 53 (103) | 5 (1) | 34 (65) |
| Algerian | Chl$_{surf}$ ($\mu$g l$^{-1}$) | 0.35 (0.08) | 0.05 (0.02) | 0.17 (0.03) | 0.76 (0.15) | 0.33 (0.07) |
| | Chl$_{max}$ ($\mu$g l$^{-1}$) | 1.00 (0.39) | 0.95 (0.45) | 0.90 (0.32) | 0.80 (0.18) | 0.92 (0.34) |
| | max depth (m) | 52 (88) | 78 (117) | 55 (124) | 14 (53) | 50 (96) |
| Tyrrhenian | Chl$_{surf}$ ($\mu$g l$^{-1}$) | 0.27 (0.07) | 0.03 (0.02) | 0.13 (0.03) | 0.51 (0.12) | 0.24 (0.06) |
| | Chl$_{max}$ ($\mu$g l$^{-1}$) | 0.92 (0.35) | 1.20 (0.38) | 0.80 (0.29) | 0.55 (0.16) | 0.87 (0.29) |
| | max depth (m) | 60 (98) | 75 (117) | 67 (131) | 23 (63) | 56 (102) |
| Ionian | Chl$_{surf}$ ($\mu$g l$^{-1}$) | 0.18 (0.11) | 0.04 (0.02) | 0.17 (0.03) | 0.33 (0.14) | 0.18 (0.07) |
| | Chl$_{max}$ ($\mu$g l$^{-1}$) | 0.54 (0.26) | 0.52 (0.30) | 0.42 (0.24) | 0.35 (0.15) | 0.46 (0.24) |
| | max depth (m) | 72 (74) | 90 (139) | 75 (156) | 16 (49) | 63 (105) |
| Levantine | Chl$_{surf}$ ($\mu$g l$^{-1}$) | 0.05 (0.07) | 0.02 (0.04) | 0.08 (0.05) | 0.22 (0.10) | 0.09 (0.06) |
| | Chl$_{max}$ ($\mu$g l$^{-1}$) | 0.41 (0.17) | 0.43 (0.22) | 0.34 (0.20) | 0.24 (0.11) | 0.36 (0.17) |
| | max depth (m) | 97 (139) | 123 (186) | 110 (186) | 47 (114) | 94 (156) |
| Western regions | Chl$_{surf}$ ($\mu$g l$^{-1}$) | 0.82 (0.14) | 0.07 (0.03) | 0.18 (0.05) | 0.67 (0.21) | 0.44 (0.11) |
| | Chl$_{max}$ ($\mu$g l$^{-1}$) | 1.36 (0.42) | 1.06 (0.52) | 0.76 (0.36) | 0.69 (0.23) | 0.97 (0.38) |
| | max depth (m) | 41 (72) | 64 (108) | 54 (114) | 10 (27) | 42 (80) |
| Eastern regions | Chl$_{surf}$ ($\mu$g l$^{-1}$) | 0.17 (0.08) | 0.03 (0.03) | 0.13 (0.04) | 0.35 (0.12) | 0.17 (0.07) |
| | Chl$_{max}$ ($\mu$g l$^{-1}$) | 0.62 (0.26) | 0.72 (0.30) | 0.52 (0.24) | 0.38 (0.14) | 0.56 (0.24) |
| | max depth (m) | 76 (104) | 96 (147) | 84 (158) | 29 (75) | 71 (121) |
| All regions | Chl$_{surf}$ ($\mu$g l$^{-1}$) | 0.43 (0.10) | 0.05 (0.03) | 0.15 (0.04) | 0.48 (0.16) | 0.28 (0.08) |
| | Chl$_{max}$ ($\mu$g l$^{-1}$) | 0.92 (0.32) | 0.86 (0.39) | 0.62 (0.29) | 0.50 (0.17) | 0.72 (0.29) |
| | max depth (m) | 62 (91) | 83 (131) | 72 (140) | 21 (56) | 60 (105) |





**Table 2.** Stability test results for model and satellite surface chlorophyll based clusters. Results are here for 4 and 5 clusters. The stability tests are done using Hennig (2007) method (see the text for more details).

|  | Satellite | | | | | PISCES | | | | |
|---|---|---|---|---|---|---|---|---|---|---|
|  | CL1 | CL2 | CL3 | CL4 | CL5 | CL1 | CL2 | CL3 | CL4 | CL5 |
| Bootstrap : | 0.89 | 0.94 | 0.90 | 0.93 | - | 0.98 | 0.99 | 0.99 | 0.99 | - |
| Noise : | 0.87 | 0.93 | 0.83 | 0.97 | - | 0.93 | 0.95 | 0.94 | 0.96 | - |
| Jitter : | 0.94 | 0.97 | 0.94 | 0.95 | - | 1 | 1 | 1 | 1 | - |
| Bootstrap : | 0.97 | 0.97 | 0.98 | 0.98 | 0.98 | 0.92 | 0.75 | 0.80 | 0.86 | 0.90 |
| Noise : | 0.91 | 0.90 | 0.95 | 0.96 | 0.87 | 0.88 | 0.64 | 0.72 | 0.84 | 0.84 |
| Jitter : | 1 | 1 | 1 | 1 | 1 | 0.93 | 0.80 | 0.84 | 0.88 | 0.93 |

**Table 3.** Stability test results for model total chlorophyll (sum of Chl biomass between 0-300m depth) and total phytoplankton (sum of phytoplankton biomass between 0-300m depth) based clusters. Results are here for 4 and 5 clusters. The stability tests are done using Hennig (2007) method (see the text for more details).

|  | $Chl_{tot}$ | | | | | phyto. biomass | | | | |
|---|---|---|---|---|---|---|---|---|---|---|
|  | CL1 | CL2 | CL3 | CL4 | CL5 | CL1 | CL2 | CL3 | CL4 | CL5 |
| Bootstrap : | 0.99 | 0.99 | 0.98 | 0.98 | - | 0.99 | 0.99 | 0.98 | 0.98 | - |
| Noise : | 0.85 | 0.94 | 0.83 | 0.92 | - | 0.95 | 0.81 | 0.84 | 0.87 | - |
| Jitter : | 1 | 1 | 1 | 1 | - | 1 | 1 | 1 | 1 | - |
| Bootstrap : | 0.96 | 0.96 | 0.99 | 0.98 | 0.98 | 0.96 | 0.92 | 0.95 | 0.92 | 0.99 |
| Noise : | 0.81 | 0.73 | 0.90 | 0.91 | 0.78 | 0.84 | 0.74 | 0.74 | 0.73 | 0.89 |
| Jitter : | 0.97 | 0.98 | 1 | 0.99 | 1 | 1 | 1 | 1 | 1 | 1 |

**Table 4.** Stability test results for model maximum chlorophyll (layer where the maximum of Chl is found on the water-column - see Fig. 7) based clusters. Results are here for 2 and 3 clusters. The stability tests are done using Hennig (2007) method (see the text for more details).

|  | $Chl_{max}$ | | |
|---|---|---|---|
|  | CL1 | CL2 | CL3 |
| Bootstrap : | 1 | 1 | - |
| Noise : | 1 | 1 | - |
| Jitter : | 1 | 1 | - |
| Bootstrap : | 0.65 | 0.83 | 0.74 |
| Noise : | 0.50 | 0.74 | 0.65 |
| Jitter : | 0.65 | 0.83 | 0.77 |





**Table 5.** Stability test results for model primary production based clusters. Results are here for 3,4 and 5 clusters. The stability tests are done using Hennig (2007) method (see the text for more details).

| | Primary Prod. | | | | |
| --- | --- | --- | --- | --- | --- |
| | CL1 | CL2 | CL3 | | |
| Bootstrap : | 1 | 0.99 | 1 | - | - |
| Noise : | 0.97 | 0.91 | 0.97 | - | - |
| Jitter : | 1 | 1 | 1 | - | - |
| Bootstrap : | 0.76 | 0.97 | 0.85 | 0.93 | - |
| Noise : | 0.40 | 0.88 | 0.61 | 0.80 | - |
| Jitter : | 0.67 | 0.95 | 0.79 | 0.92 | - |
| Bootstrap : | 0.96 | 0.89 | 0.97 | 0.90 | 0.83 |
| Noise : | 0.63 | 0.70 | 0.86 | 0.73 | 0.64 |
| Jitter : | 0.97 | 0.97 | 0.98 | 0.97 | 0.95 |



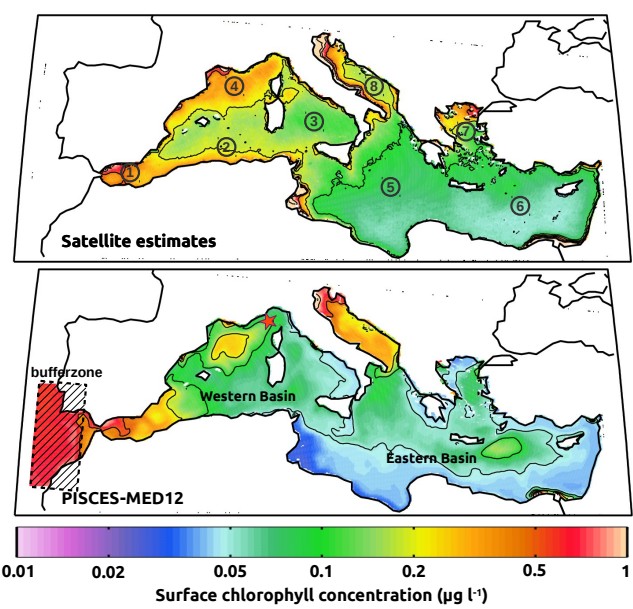

**Figure 1.** Surface chlorophyll concentrations estimated (upper caption) from remote sensing using the Mediterranean specific method of Bosc et al. (2004), and (lower caption) by the PISCES-MED12 model(presenting at the same time the MED12 domain). Both chlorophyll fields are annual averages based on the same 8 years period (Nov. 1997 - Oct. 2005). The Mediterranean sub-basins are also presented. The Western Mediterranean sub-basins: 1- Alboran, 2- Algerian, 3- Tyrrhenian, 4- Liguro-Provençal; and those from the Eastern basin: 5- Ionian, 6- Levantine, 7- Aegean, and 8- Adriatic. The red star in the Liguro-Provençal sub-basin (just North of Corsica) indicates the location of the DYFAMED station.





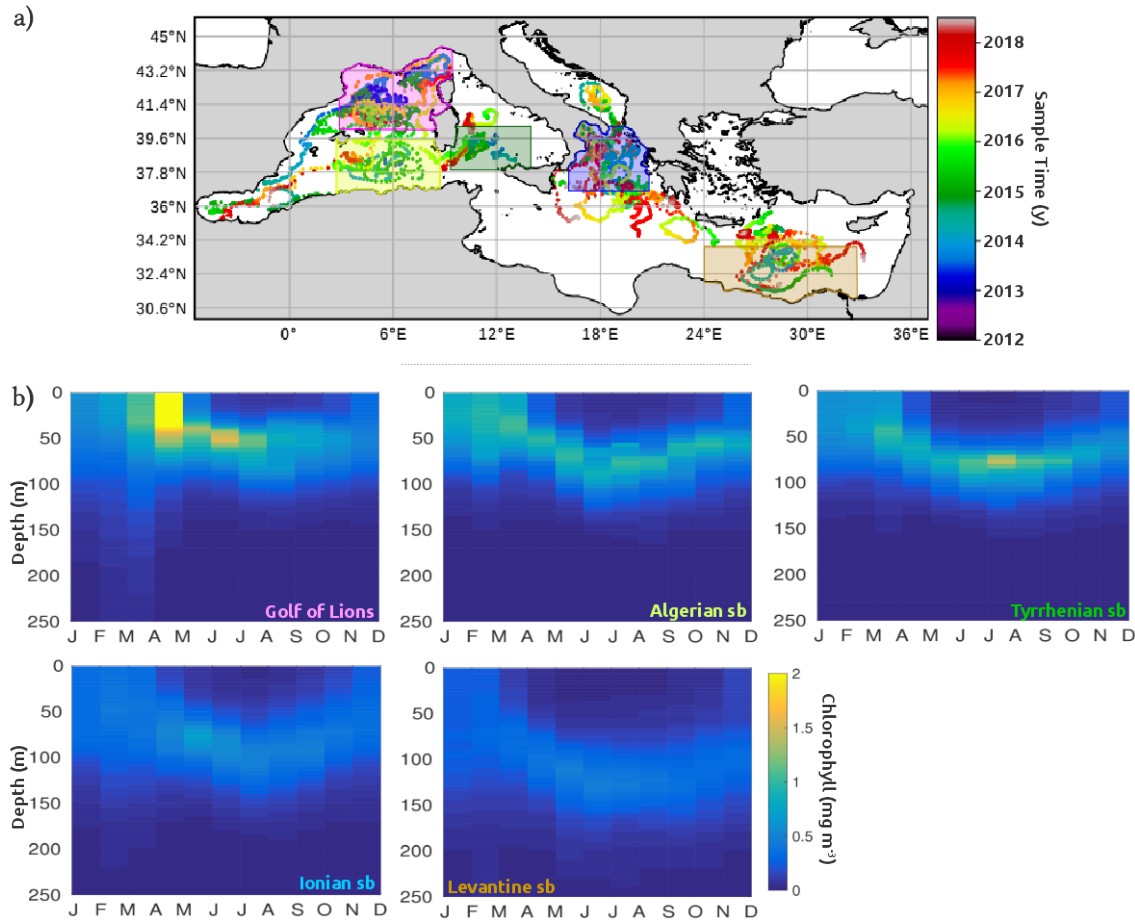

**Figure 2.** Presentation of the *Argo-Bio* floats of the Mediterranean sea. (See http://www.obs-vlfr.fr/~dortenzio/naos_wp3/index.html and http://www.oao.obs-vlfr.fr/maps/en/). *Argo-Bio* floats are transported by the current like a lagrangian particle, sampling vertical profiles down to 1000m depth, every 8 days, of : temperature, salinity, nitrate, dissolved oxygen, chlorophyll, Photosynthetic Active Radiation (PAR), Coloured Dissolved Organic Carbon (CDOM), and particulate optical backscattering (bbp). a− Map of all 8-days float sample positions in the Mediterranean sea, and the date of all samples (colour). 5 regions are also defined on the map: Liguro-Provençal (pink), Algerian (yellow), Tyrrhenian (green), North Ionian (blue), and South Levantine (brown). b− Monthly mean progression of the *Argo-Bio* vertical chlorophyll profile, in the different regions presented just above.





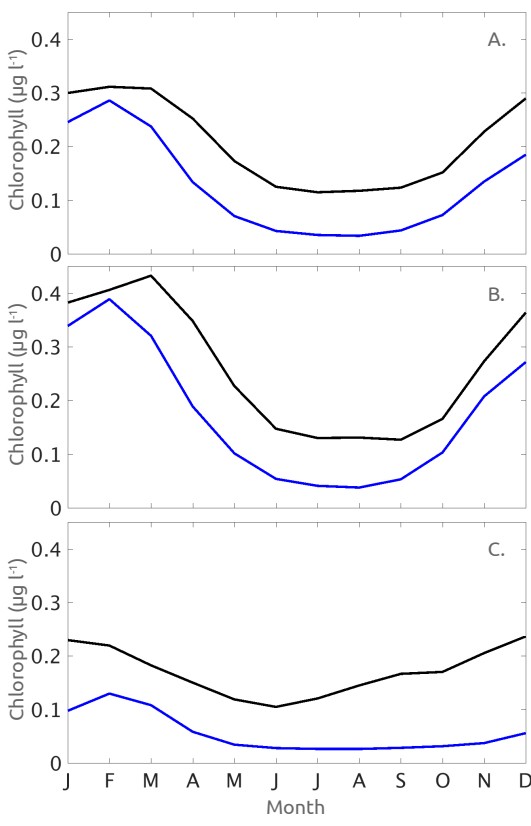

**Figure 3.** Temporal evolution of model (blue) and satellite estimated (black line; estimates from Bosc et al. (2004)) $Chl_{surf}$, monthly averaged evolution from time period covered by the satellite estimates (1997-2005). The upper pictures (A) include the whole Mediterranean Sea, in the middle (B) the Western basin, and down (C) the Eastern basin (Levantine and Ionian sub-basin).





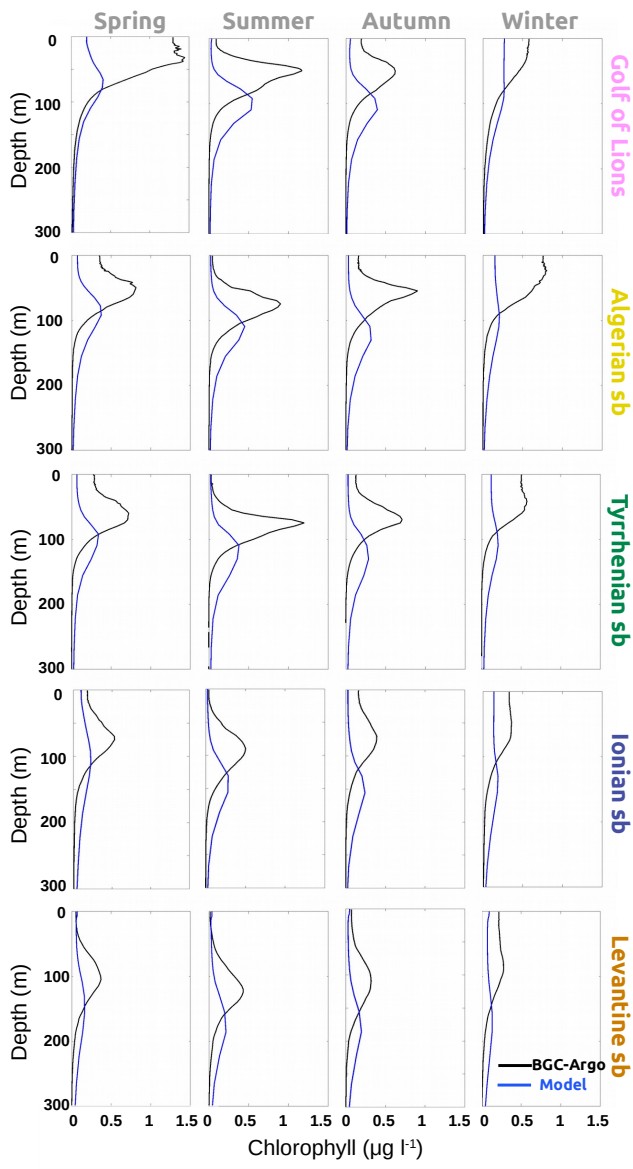

**Figure 4.** Seasonal averaged vertical profiles of chlorophyll concentration at the different regions defined on picture 2, from *BGC-Argo in situ* observations (black) and model results (blue).





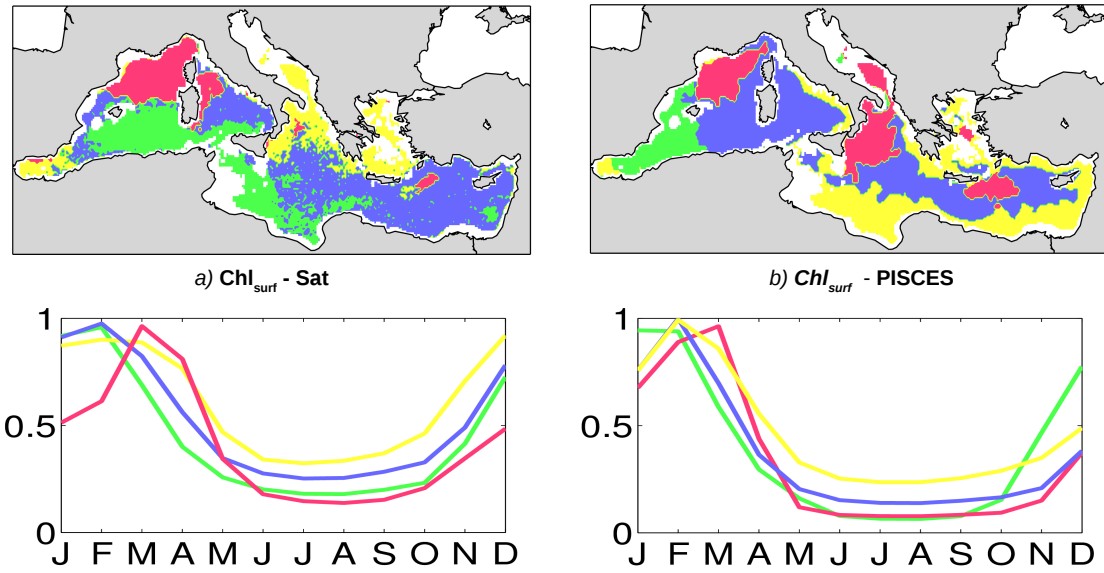

**Figure 5.** Mapping of the different clusters resulting of k-mean exercise on Mediterranean surface chlorophyll, corresponding to the different bioregions of the Mediterranean Sea (up) and their respective normalized annual cycle (down) for (a) remote sensing chlorophyll fields from Bosc et al. (2004), and (b) PISCES-MED12 model.

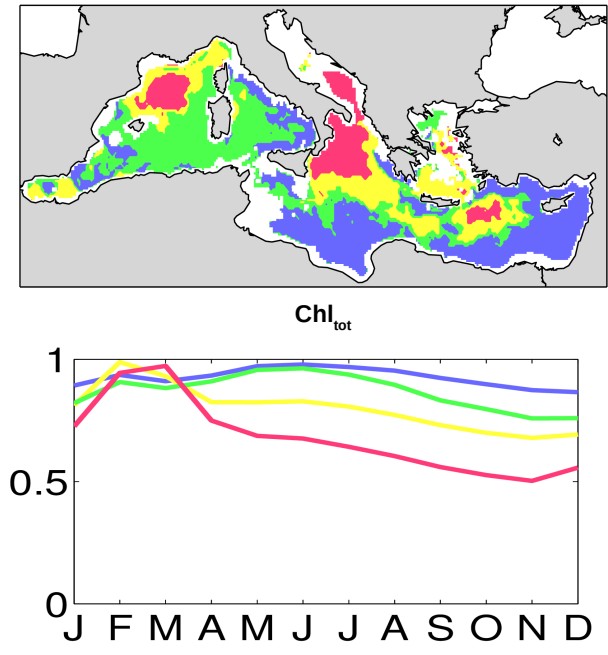

**Figure 6.** Same as Fig. 5, but applied to the sum of chlorophyll on the 0 - 300 m depth layer.





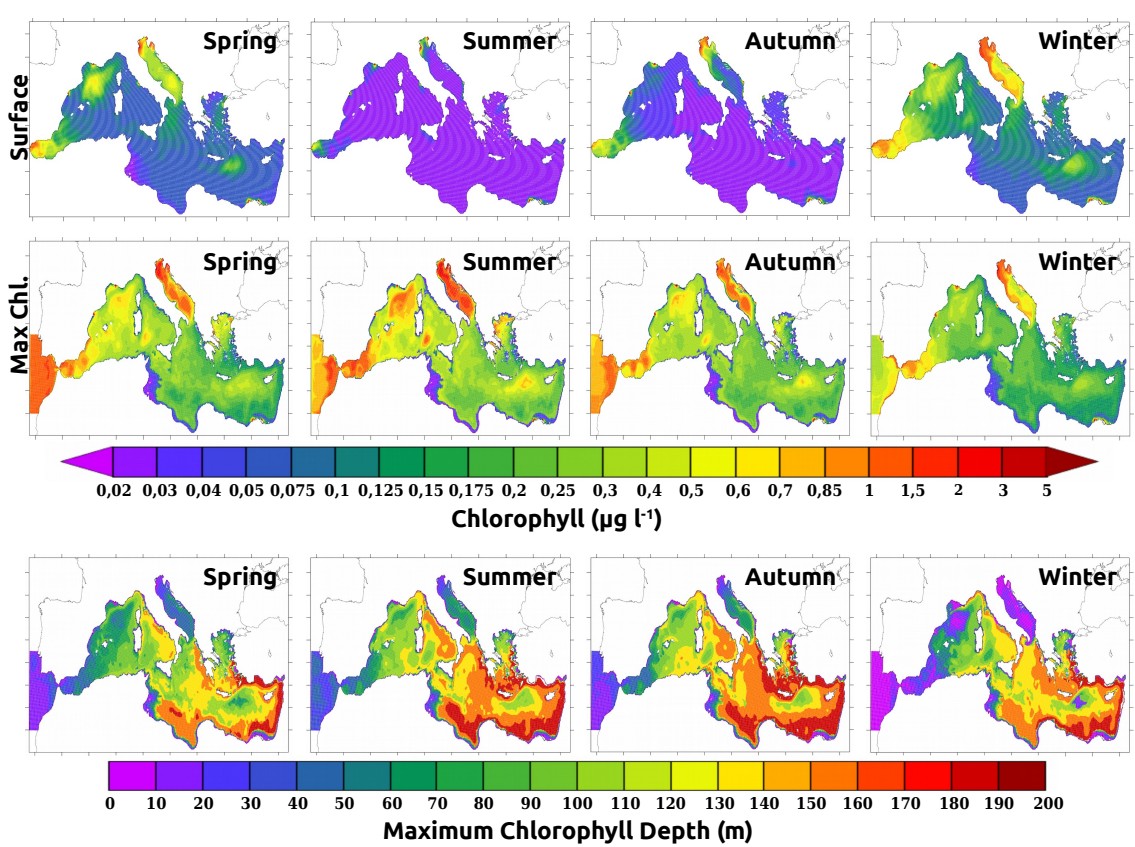

**Figure 7.** Seasonal comparison of the model surface chlorophyll fields (in $\mu$g l$^{-1}$ ; 10 first meters average)(up), and of the model chlorophyll maximum found on the water column (middle), with the corresponding depth of this chlorophyll maximum (in meter ; down).



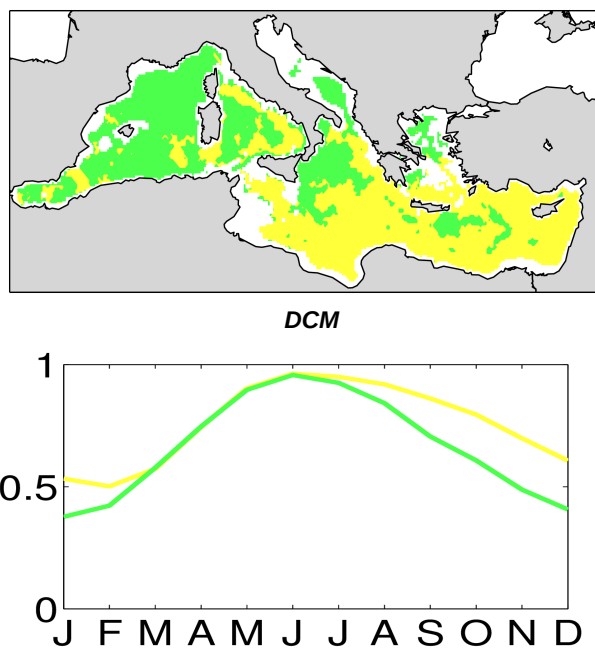

**Figure 8.** Same as Fig. 5, but applied to the maximum chlorophyll layer.

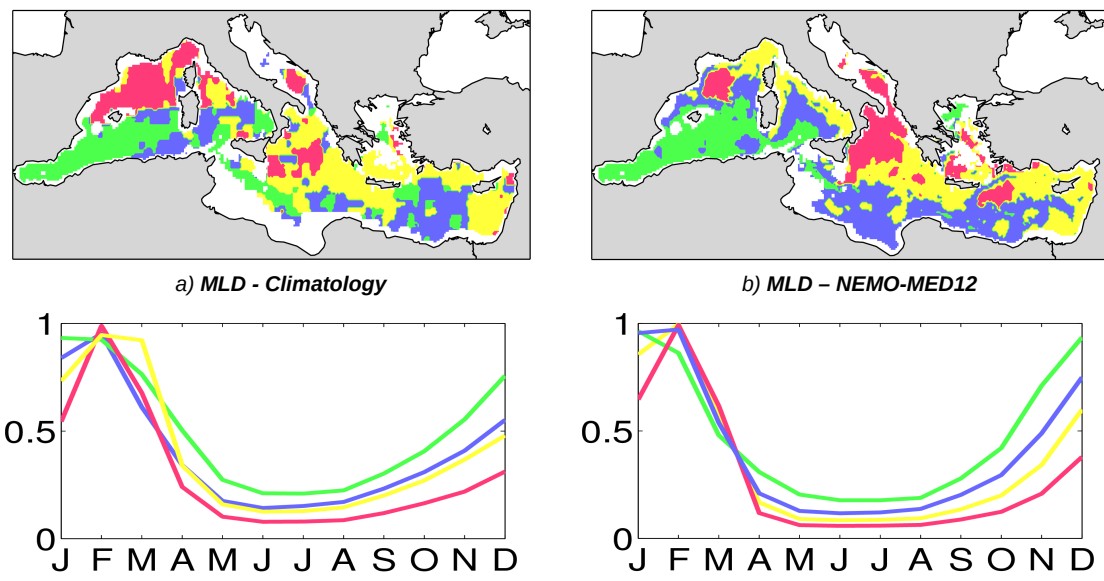

**Figure 9.** Same as Fig. 5, but applied to the Mixed Layer Depth (MLD) coming from (a) The new Mediterranean MLD climatology of Houpert et al. (2015), and (b) NEMO-MED12 dynamical fields.





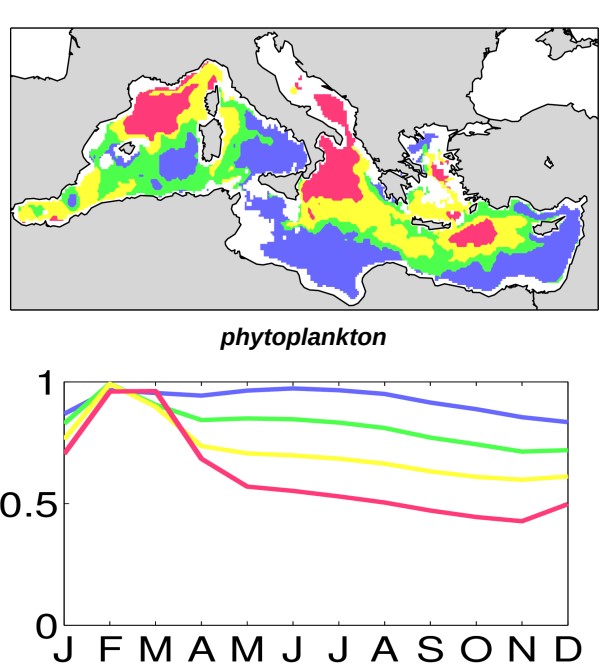

**Figure 10.** Same as Fig. 5, but applied to the sum of phytoplankton biomass on the $0 - 300$ m layer.





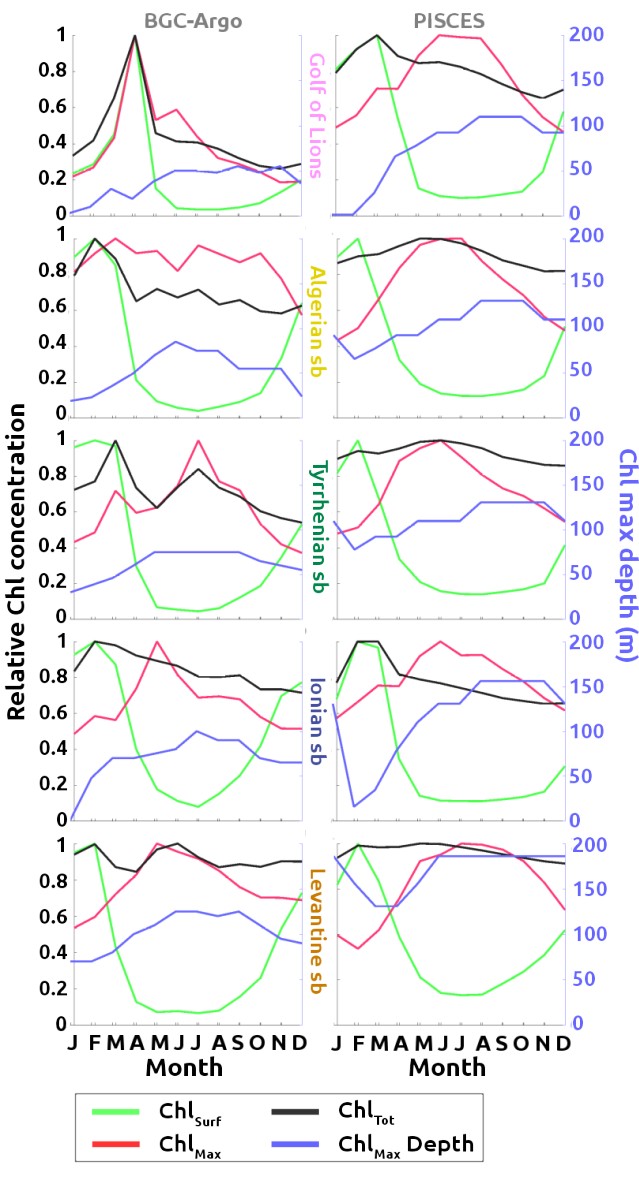

**Figure 11.** Normalized annual cycle of $Chl_{surf}$ (green line), $Chl_{max}$ (red line), $Chl_{tot}$ (black line), and of the maximum chlorophyll depth (blue line), from the *BGC-Argo* floats data (left column) and the PISCES model (right column), in the 5 areas defined in the figure 2.

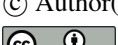



## Appendix A:  PISCES low surface chlorophyll

This PISCES-MED12 configuration is not able to model high Chl concentration at low depth (figure 4, table 1). The main reason of this problem is a strong $PO_4$ limitation on the first 100m all over the Mediterranean sea (Figure A2), in particular in the western basin.

One reason of this problem could be the PISCES C/N/P Redfield ratio in organic matter production and remineralization. This fix ratio does not enable different remineralization speed for organic carbon, nitrogen or phosphorus. Different remineralization speeds are extremely important, in particular in phosphate limited area like the Mediterranean sea, where organic phosphates are well known to be degraded in priority (Thingstad and Rassoulzadegan, 1995; Thingstad et al., 1996; Moutin, 2000; Moutin et al., 2002; Pujo-Pay et al., 2011). Guyennon et al. (2015) have done a "twin" simulation, with the same circulation fields, the same external inputs, same initial fields, and the ECO3M biogeochemical model (Baklouti et al., 2006a, b) instead of PISCES. ECO3M aims to model processes at cell level, and is a "non-Redfieldian" model, that enable different C/N/P ratio in the organic matter, and hence different remineralization speed for the organic carbon, nitrogen and phosphorus. But their simulation resulted in a similar smooth nutricline, and a strong phosphate under-estimates over the first 100m depth. So PISCES fixed Redfield ratio does not explain the surface $PO_4$ limitation of this run, at least not alone.

Another explanation could come from the circulation model dynamics. The strong − missing − Western Mediterranean nutricline is located in the intermediate water (Figure A2). NEMO-MED12 circulation evaluation with CFC (Palmiéri et al., 2015; Palmiéri, 2014), and helium3/tritium (Ayache et al., 2015) model/data comparisons, show an over-ventilation of the intermediate water, due to (i) a too strong LIW formation in the Levantine sub-basin, and (ii) a too strong mixing in the Northern Ionian (what is also visible through the red MLD cluster on the figure 9). The impact of this over-ventilation on the model biogeochemistry is visible on the intermediate waters Apparent Oxygen Use (AOU; Figure A1). PISCES-MED12 globally under-estimates the Mediterranean intermediate water AOU of $\sim$30 $\mu$mol l$^{-1}$. AOU progression however is almost the same in model and observation from the Northern Ionian to the Algerian sub-basin (from 35 to 65-70 $\mu$mol l$^{-1}$ in MEDATLAS and from 5-10 to 35 $\mu$mol l$^{-1}$ in PISCES), The remineralization rate is then correct in PISCES-MED12, but the intermediate waters are too "young" in the model. Hence its organic matter remineralization cannot be advanced enough to create the strong and shallow Western Mediterranean nutricline, and explains (at least partially) why both PISCES and ECO3m were not able to model it.

Recently, Richon et al. (2018c) performed a new PISCES-MED12 simulation, with the same PISCES-MED12 model version than used in this study, but forced by new NEMO-MED12 dynamics Hamon et al. (2016). These new dynamics result in an improved intermediate water ventilation (Ayache et al., 2016), which slightly improves the nutricline, even if still too smooth (especially in the Western basin). Resulting DCM is this run is much better, both DCM value and depth is really improved compare to the run used in this study. But because of a too stable eddy in the Western basin, the surface chlorophyll dynamics of this run are too flat: summer minimum is equivalent to remote sensing estimates, but winter/spring maximum is way too low, and the Gulf of Lions spring bloom bioregion was even not present after clusterization (not shown). Despite undeniable improvements in Richon et al. (2018c) simulation, we decided to not use it in this study, because the $Chl_{surf}$ to $Chl_{max}$



contribution to $Chl_{tot}$ was too unbalanced compare to observations, to perform correct $Chl_{tot}$ analysis. This simulation shows us anyway that the improved intermediate water ventilation helps improving the nutricline, which in turn improved the DCM concentration and depth. It might also helped increasing the surface Chl minimum in summer, but that run also includes atmosphere deposition what is more likely to be the determinant factor that boosted the summer surface primary production

5   (as shown in Richon et al. (2018c)). Both simulations (this study and Richon et al. (2018c)) present interesting differences, that would benefit from more detailed comparison.





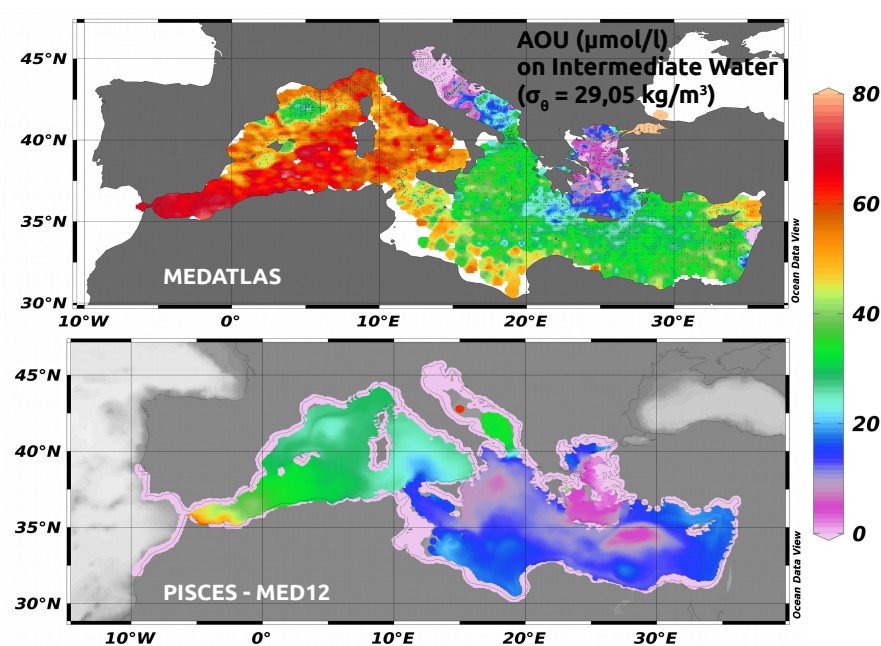

**Figure A1.** Comparison of the Apparent Oxygen Use (AOU) change in $\mu$mol l$^{-1}$ along the intermediate water of the Mediterranean sea, using observations from the MEDAR/MEDATLAS climatology (MEDAR/MEDATLAS-Group, 2002), and the model results from the 1995-2004 decadal average. The intermediate waters were isolated by following their specific isopycnal surface: $\sigma_\theta$=29.05 (Roether et al., 1998).



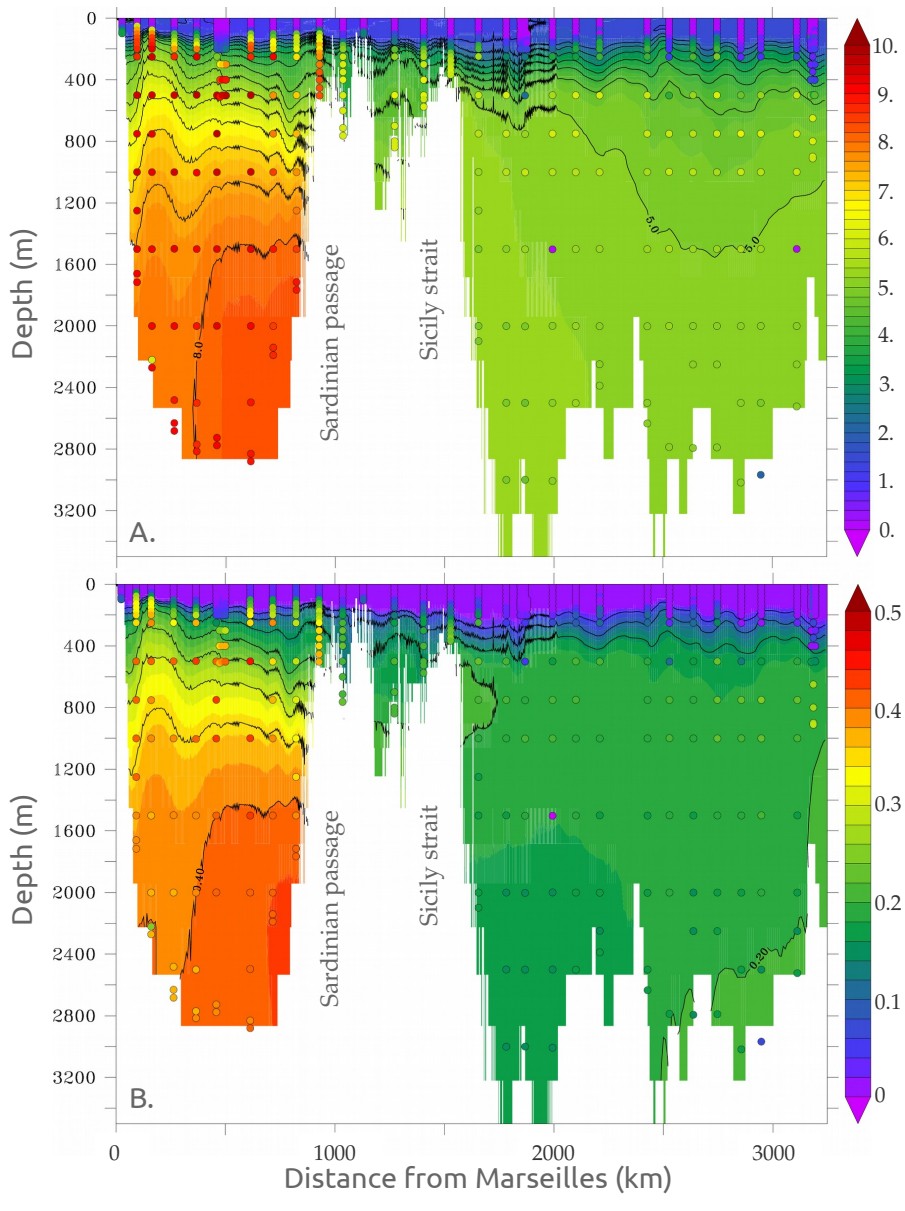

**Figure A2.** NO$_3$ (A.) and PO$_4$ (B.) concentration in $\mu$mol l$^{-1}$ along the BOUM cruise section (Moutin et al., 2012). Colored dots are *in situ* concentration, while the font colors are from the PISCES model results, for the same time period than the BOUM cruise (August 2008).



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
