# Peer review of "The Mediterranean subsurface phytoplankton dynamics and their impact on Mediterranean bioregions"

_Biogeosciences, 2018_

## Referee Comment (RC1) · Anonymous Referee #1 · 21 Nov 2018

Overall assessment

In this work, the authors use a 3D coupled hydrodynamic-biogeochemical model of the entire Mediterranean basin to investigate the dynamics of subsurface chlorophyll accumulations and its relative contribution to total plankton production. They compare surface model results and satellite chlorophyll estimates (also from the surface) with integrated ( 300m) simulated phytoplankton values and conclude that the contribution from the subsurface levels is important and provides a very different picture of the usually accepted 'oligotrophic' Mediterranean Sea. Model simulations are also assessed against the newly available bio-ARGO data to understand to which extend simulated

vertical chlorophyll values match with field data.

The topic here studied is highly relevant for our understanding of functioning of the Mediterranean Sea ecosystem and also to better evaluate the generic oceanographic knowledge usually provided by surface-only information, as the one obtained from remote sensing. However, and as much as I liked the topic and approach used, I have severe concerns about the suitability of the used model to address the scientific questions being asked in this work. The authors have tried hard to overcome the obvious limitations of the model which, on the other hand, is common to ALL modelling approaches but I still have some concerns as detailed on the following paragraphs.

General comments:

My major concern regards the lack of concordance between simulated and measured chlorophyll values. First for the surface chlorophyll values. From the map in Fig. 1 it could be quite obviously seen that mean simulated surface chlorophyll values are much lower almost everywhere than satellite (even if the chosen color-scale makes the comparison a bit hard). This is confirmed by the seasonal cycles shown in Fig. 3 where the sub-estimation of chlorophyll by the model at surface is plain for all investigated sites (as the authors state: model surface Chl globally under-estimates satellite values by a factor 2). In the following paragraph this difference is partially justified by known biases in satellite estimates for the Mediterranean Sea but I am totally sure that satellite information is not that far away from field measurements.

Then, this sub-estimation of chlorophyll levels is also observed for the deep structures. In this case the comparison between model and bio-ARGO data shows '..that the model underestimates the Chl concentration, not only at the surface, but also at depth by 60Further, from the comparison made in Fig. 11 it is quite clear than not even the relative chlorophyll (with respect to its maximum monthly value) is properly simulated by the model (at least for the western Mediterranean regions).

These deviations commented above are, in my opinion, large enough to prevent using

the model for the intended analysis on the chlorophyll phenology. I appreciate the effort made by the authors to make the comparison model/data quantitative and to provide hypothesis on why the model fails to reproduce observed patterns. As stated in Appendix, the too-deep nutricline (especially for phosphate) seems to be the reason of the observed differences. Either the use of fixed internal nutrient ratios or (more likely) hydrodynamic model deficiencies being the causes of the miss-matches. The fact that another biogeochemical model coupled to the same hydrodynamic data improves the DCM simulation but worsens the surface conditions make me wonder if maybe NEMO (at least in the current configuration) is an appropriate choice for making biogeochemical simulations in the Mediterranean Sea. I am aware this model is being widely used in this basin but the results shown in this submission are somehow worrisome, at least when it comes to simulate the biogeochemistry.

I have also some minors comments on the wording of the manuscript, especially when considering the model/data deviation (as the authors are overly optimistic in my opinion) but I am not providing them in here as unless the authors could generate a new simulation in which the basic characteristic of the DCM (and of the chlorophyll in general) are better aligned with the observations I sincerely doubt that this model could be used for the objectives presented in the manuscript.

---

## Author Comment (AC1) · 24 Nov 2018

Although the referee #1 appreciates the analyses, the idea, and the importance of the subject discussed in the paper, he/she doesn't think the model used is appropriate for this study. Mainly the chlorophyll performances are thought to be too poor for this analysis, and the NEMO model is maybe not appropriate to model the Mediterranean sea. We will discuss this in the following paragraphs, and try to convince that even though the model chlorophyll has imperfections, we think it is appropriate to perform our analysis. Our study provides interesting analysis on the classification of the bioregions in the Mediterranean basin, which we think are useful to provide to our scientific community.

[Figure]

* About NEMO in the Mediterranean Sea.

Modelling the Mediterranean Sea circulation requests a high resolution model with adapted atmospheric forcing that includes its surrounding topography, to get the right winds that will induce specific eddies and deep water mass formation. It is not an easy task, especially without any kind of data assimilation or surface relaxation, and still the NEMO-MED12 configuration manage to model probably the best (whole) Mediterranean circulation. It includes specific events like the Eastern and Western Mediterranean Transient, and is probably the most evaluated regional circulation configuration, with not only dynamic evaluation (Temperature, Salinity), but also transient passive tracer that track the sea interior ventilation (CFC, tritium, . . ., see Palmiéri et al. (2015), Ayache et al. (2015), Ayache et al. (2016)).

Of course It is not perfect, no model is, but because it is well evaluated, we managed to highlight the model strengths and weaknesses, document the improvements between each version (better intermediate water ventilation when improving the atmospheric forcing, better deep water circulation when changing from 50 to 75 vertical levels, ...), and it helps understanding the model circulation impact on the biogeochemistry (this is unique in the Mediterranean modelling community).

So, we can discuss how appropriate NEMO is to model the Mediterranean sea, but we should also admit that the teams behind these NEMO-MED configurations are doing a good and healthy job, and obtain nice results.

* About our modelled chlorophyll.

- Pisces is a biogeochemical model. We mainly focussed the discussion on the chlorophyll for our analysis, but we should not forget all the other variables modelled, that are reasonably correct (See Fig. A2 for nutrients, and Fig. 1 for surface Chl), so that it is appropriated to conduct interesting biogeochemical studies. Our model results are sensible, it captures the main features of the Mediterranean sea, and other biogeochemical models are doing similar jobs (Lazzari et al. (2012), Mattia et al. (2013),

Guyennon et al. (2015), Macias et al. (2014)). If you look all these other Mediterranean modelling studies, we do evaluate our model performances much more than it is usually done. This is because we think it is good practice to not skip or hide the model weaknesses, but rather to show and discuss them: 1- To help finding the origin of the problems, we share the informations so that we can discuss solutions with other groups, 2- Because it gives a good overview, a better understanding of our model to the reader, and we think it is important. So yes, the model chlorophyll does not perfectly match the observations, and you know why it could be so. Moreover, a similar version of the NEMO/PISCES model, not perfect as well, has already been used for published biogeochemical investigation in the Mediterranean basin (Richon et al, 2018a,b)

- We especially used BGC-Argo observations (We are probably the first modelling group to evaluate our model with these data) to show that surface, max and total chlorophyll patterns found in the model are present in observations. This is a key point. I would agree that we could not easily believe the model integrated chlorophyll phenology and derived bioregions only based on the vertical profile comparisons (Fig. 4). But the vertically-integrated chlorophyll phenologies from the BGC-Argo show similar patterns (Fig. 11), with the winter and the summer chlorophyll maxima! And this (maybe we should make it clearer in the text) demonstrates that even if the model chlorophyll has weaknesses, our analysis is sensible, and the conclusions drawn from it actually make sense. A model will never be perfect in all aspects, and it is so regrettable to be penalized because we provide more evaluation than usually performed.

Finally, as the referee said, "the topic is highly relevant". We strongly think that our conclusions drawn from the model and supported by observations are realistic, and we are convinced it is an interesting study that should be useful to the wider community.

References:

Ayache, M., Dutay, J.-C, Jean-Baptiste, P., Beranger, K., Arsouze, T., Beuvier, J., Palmieri, J., Le-vu, B., and Roether, W.: Modelling of the anthropogenic tritium transient and its decay product helium-3 in the Mediterranean Sea using a high-resolution regional model, Ocean Science, 11, 323–342, https://doi.org/10.5194/os-11-323-2015, 2015

Ayache, M., Dutay, J.-C., Arsouze, T., Révillon, S., Beuvier, J., and Jeandel, C.: High-resolution neodymium characterization along the Mediterranean margins and modelling of $\varepsilon$Nd distribution in the Mediterranean basins, Biogeosciences, 13, 5259–5276, https://doi.org/10.5194/bg-13-5259-2016, https://www.biogeosciences.net/13/5259/2016/, 2016.

Guyennon, A., Baklouti, M., Diaz, F., Palmieri, J., Beuvier, J., Lebaupin-Brossier, C., Arsouze, T., Béranger, K., Dutay, J.-C., and Moutin, T.: New insights into the organic carbon export in the Mediterranean Sea from 3-D modeling, Biogeosciences, 12, 7025–7046, https://doi.org/10.5194/bg-12-7025-2015, 2015.

Lazzari, P., Solidoro, C., Ibello, V., Salon, S., Teruzzi, A., Béranger, K., Colella, S., and Crise, A.: Seasonal and interannual variability of plankton chlorophyll and primary production in the Mediterranean Sea : a modelling approach, Biogeosciences Discussions, 9(1), 217–233, DOI: 10.5194/bg–9–217–2012, 2012

Macías, D., Stips, A., and Garcia-Gorriz, E.: The relevance of deep chlorophyll maximum in the open Mediterranean Sea evaluated through 3D hydrodynamic-biogeochemical coupled simulations, Ecological Modelling, 281, 26–37, 2014.

Mattia, G., M. Zavatarelli, M. Vichi, and P. Oddo, The Eastern Mediterranean Sea biogeochemical dynamics in the 1990s: A numerical study, J. Geophys. Res. Oceans, 118, 2231–2248, doi:10.1002/jgrc.20160, 2013

Palmiéri, J., Orr, J. C., Dutay, J.-C., Béranger, K., Schneider, A., Beuvier, J., and Somot, S.: Simulated anthropogenic $CO_2$ storage and acidification of the Mediterranean Sea, Biogeosciences, 12, 781–802, https://doi.org/10.5194/bg-12-781-2015, http://www.biogeosciences.net/12/781/2015/, 2015.

[Figure]

Richon C., J-C Dutay, F Dulac, R Wang, and Y Balkanski, Modeling the biogeochemical impact of atmospheric phosphate deposition from desert dust and combustion sources to the Mediterranean Sea, Biogeosciences, 15, 2499–2524, 2018

Richon C, J-C Dutay J-C, F. Dulac, P. Nabat, R. Wang, Y. Balkanski, O. Aumont, C. Guieu, K. Desboeufs, B. Laurent, P. Raimbault, and J. Beuvier, Modeling atmospheric deposition impacts on major nutrients and biological budgets of the Mediterranean Sea, 2018,, Progress in Oceanography,163, 21-39, , doi.org/10.5194/bg-15-2499-2018

---

## Referee Comment (RC2) · Anonymous Referee #1 · 28 Nov 2018

The authors wrote in their reply to my first comments:

'..he/she doesn't think the model used is appropriate for this study...and the NEMO model is maybe not appropriate to model the Mediterranean sea.'

I'm sorry if the authors misunderstood my previous review and the comment about NEMO. I wrote that '. . .make me wonder if maybe NEMO (at least in the current configuration) is an appropriate choice for making biogeochemical simulations in the Mediterranean Sea'. So what I am unsure is whether the NEMO configuration the authors are using is appropriate to perform biogeochemical simulations, given the already reported problems with vertical stratification strength and, hence, with the position of the nutricline.

I fully agree in NEMO being a 'nice' hydrodynamic model that has been applied and evaluated in many instances to study Mediterranean Sea characteristics. I am not questioning previous works, I am just saying that the results about biogeochemistry and, particularly, the chlorophyll distribution the authors are showing are not accurate enough to make the analysis they propose.

They also make a couple of claims in their reply which I think are not fair and are not useful to the objective at hand (i.e., evaluate their own work). At some point they say: 'If you look all these other Mediterranean modelling studies, we do evaluate our model performances much more than it is usually done.' It could be discussed whether the model validation the authors perform in the present contribution is more thorough than in other previous works but we should never defend our own work by criticizing other colleague's publications, especially when those other works passed through a similar peer-review process you are undergoing right now.

Then, in another paragraph they say: 'A model will never be perfect in all aspects, and it is so regrettable to be penalized because we provide more evaluation than usually performed'. I am, again, sorry if the authors felt unfairly 'penalized' by my previous review but I try to do an objective evaluation of every work I review, exactly as I want my papers to be reviewed. Being a modeler myself I perfectly know no model is perfect and fully appreciate the effort to compare your model results with different sets of measurements. However (and unfortunately) these comparisons show that your simulations are, simply, too far off the observations to be useful (remember you are underestimating chlorophyll by 60-70% and the position of the DCM in more than 50m!). Not even the total chlorophyll in relative terms (your figure 11) is sufficiently close to the Argo data to be useful for the analysis.

Instead of arguing with me I should rather focus to find another simulation to perform the analysis as I do believe the primary claim by the authors (i.e., the Med bioregions

will change when DCM is considered) is basically right. You only need to have the right data to show that to the community.

---

## Referee Comment (RC3) · Anonymous Referee #2 · 14 Dec 2018

This is the review of the manuscript "The Mediterranean subsurface phytoplankton dynamics and their impact on Mediterranean bioregions" by PalmieÌA̧ri et al.. In the manuscript the authors analyzed the output of a coupled dynamical-biogeochemical model, adapted to the Mediterranean basin, mainly examining how the subsurface chlorophyll distribution could improve the knowledge of the ecosystem functioning of the basin. In order to validate the model, their output are compared with satellite observations and bio-ARGO profiles. I consider the topic absolutely interesting and I appreciated the validation efforts done by the authors. Nevertheless, I have to say that the low quality of the comparison of the model with both satellite and in situ observations prevents an efficient study of the dynamics of the subsurface chlorophyll distri-

bution in view of a better comprehension of the Mediterranean ecosystem functioning. I am aware that, for the purpose of the manuscript, the authors are mainly interested to the comparison of the phenology and that a quantitative comparison may not be strictly necessary. However, in my opinion, there are several factors that need to be addressed. For example, there is a factor of 2 between satellite and model estimates of the phytoplankton chlorophyll; there is an underestimate of 60-70% of the model estimations with respect bio-ARGO observations at surface; there is an underestimation of 60% of the model chlorophyll concentration at depth with respect to the bio-ARGO profiles; the DCM depth is always deeper (between 30 and 50m) than that measured through bio-ARGO floats. Thus, the impression is that the model (or at least the used configuration) does not allow to simulate the Mediterranean conditions not only along the water column but also at surface. Observing maps in figure 1 and figure 5, it is quite evident that satellite and model are significantly different. The authors describe similarities and differences between satellite and model emphasizing a lot the few similarities and belittling the considerable differences (i.e., spatial distribution of the bloom in Ligurian Sea as well as in Rhodes Gyre, the important differences along the African coasts). These discrepancies cannot be justified only with "a known overestimation" of satellite observations especially because the used satellite dataset is dated 2004. In this respect, the scientific community have made considerable progresses over the past 14 years, and presently, this old satellite overestimation in the Mediterranean Sea has been quite well corrected. Why did the authors do not use a more recent and easily available dataset? Observing maps of figure 5, I observe many differences and some similarities; bloom-intermittently cluster is very different and the yellow cluster in the model regionalization is mostly absent in satellite bioregionalization. In figure 11 the authors analyze the annual cycle of different chlorophyll (surf, max and tot) for bio-Argo and model in some regions of the basin. I note, again, some similarities and many differences, despite the normalization of the chlorophyll values, with respect to the maximum, that should simplify the comparison. In the Gulf of Lions, bio-Argo and model show totally different results. The situation improves slightly in the Algerian and

Tyrrhenian basins where, at least, surface chlorophyll seems to show an analogous trend between bio-Argo and model, but the others chlorophyll (max and tot) continue to be different. Results associated with the Ionian and Levantine basins are closer. In general, I think the authors should be more impartial commenting results, for which they should use the same "yardstick" for the good as well as for bad ones. In conclusion, as I wrote above, I believe the topic covered in the manuscript is absolutely stimulating and interesting. I would like to encourage the authors to try to find another way (other configurations, other models or other techniques) to reconstruct the vertical chlorophyll distribution in order to obtain a better comparison with respect to other kind of observations (satellite or in situ).

---

## Referee Comment (RC4) · Anonymous Referee #3 · 17 Dec 2018

Second submission review of:

The Mediterranean subsurface phytoplankton dynamics and their impact on Mediterranean bioregions

by Julien Palmiéri, Jean-Claude Dutay, Fabrizio D'Ortenzio, Loïc Houpert, Nicolas Mayot and Laurent Bopp

Manuscript ID: BG 2018 423

General comments

This paper discusses the differences in regionalization of the Mediterranean Sea into

spatial clusters, relative to surface chlorophyll satellite data and biogeochemical model output, as well as vertical chl profiles obtained the same model and compared to ARGO float in situ data. The topic is extremely interesting, given its ecological implications of the different regions' behavior. However, I unfortunately have to recommend rejection of the paper, but with the strong encouragement to analyze well the cause of the profound discrepancies between the model and the data and then to re-submit a new manuscript. Indeed, the model - data (satellite or ARGO) differences are so pronounced that they should be tackled quantitatively and the model should be analyzed in order to ameliorate its results. Instead, in many instances, comments are only qualitative, where they should be crucially quantitative. For example with percent difference maps/profiles, etc. and less words. In sum, I feel that the model is inadequate for this use and should be thoroughly revised. See particular comments below, concerning technical aspects. Moreover, descriptions in the Discussion are very tortuous, long and difficult to understand, also due to the necessity of English correction and because very often the figures are only scantily cited in the text, so that the reader remains confused. See the conceptual comments, among the particular comments below.

Form

The English of the manuscript needs correction. I have tried to help, in the particular comment section below.

Particular comments and suggested text corrections

NOTE: hereafter "->" means "replace"

Abstract

1 Introduction

Page 2 Line 20. "principally the remote sensing" -> "e.g. principally remote sensing". Line 28. "permit the vertical profile of phytoplankton biomass to be estimated" -> "to obtain estimations of the vertical profile of phytoplankton biomass".

Page 3 Line 3. "papers" -> "studies" Line 10. "through to" -> "all the way to". Line 13. "into the" -> "in the" Line 16 "Winter" -> "winter" (check for capitals in season names, some are lowercase, some uppercase) Line 16. "In general, DCM deepen" -> "In general, the DCM deepen" Line 21. "Overall, DCM" -> "Overall, the DCM" Line 23. "serve to make" -> "make" Line 28. "at this time" -> maybe not necessary, since you say "remain", which implies "nowadays".

2 Methods

2.1 The Mediterranean biogeochemical model: PISCES-MED12

Page 4 Line 14. "2 size class". -> "2 size classes". Line 16. "nutrient proceeds" -> "nutrients proceeds". Line 25. "Gibralter" -> "Gibraltar" Line 31. "Dissolve" -> "Dissolved " Line 33. "calculated on the" -> "calculated at the"

Page 5 Line 3. "Mediterranean sea" -> "Mediterranean Sea" Please correct throughout text.

2.2 Remote sensing fields

Line 14. "Evaluation" -> "The evaluation" Line 17. "over-estimates" -> "overestimates". No need for hyphen. Line 18. "a 8 years period" -> " an 8-year period"

2.3 Biogeochemical-Argo floats

Line 27. "point to point" -> "point-to-point" or "pointwise" Line 28. "where 1)" -> "because 1)" Line 28. "there is enough data" -> "there are enough data" Line 30. "Liguro-Provençal" -> "the Liguro-Provençal...2- the Algerian... etc.". Add "the" to all items.

2.4 Bioregionalization

Line 8. "filtering used model" -> "the filtering used the model" Line 12. "same procedure apply" -> "the same procedure applies" Line 22. "of cluster" -> " of clusters" Line 23.

"disturbing method" -> "disturbing methods" Line 25. "both disturbed" -> "both the disturbed" Line 32. "(Chlsurf > 1 ug l-1)" should this not be Chlsat?

3 Results

3.1 Model Surface and subsurface chlorophyll evaluation. Eliminate full stop.

Line 30. "both satellite-estimated and model" -> "both satellite estimates and model"

Line 1. "Gulf of Lion" -> "Gulf of Lions" (plural in English). Please search and correct throughout text. Line 2. "Maghrebin coast" -> "Maghreb coast" search and replace throughout text, please. Line 4. "produced" -> "reproduced" Line 5. "satellites estimates" -> "satellite estimates".

Conceptual comment Line 6. "elevated surface Chl values in the Eastern basin". I have a problem concerning the model result (Fig 1b), in that I see, besides the Maghreb coast underestimated chl with respect to satellite chl., i.e.:

1) the central-southern Adriatic Sea chl is much higher that the satellite estimate 2) The Adriatic Sea cross-shore gradients are much weaker in the simulations, as if the coastal (chl-rich) current were absent 4) the southern Eastern basin simulated chl is quite lower than the sat chl 5) the North Aegean simulated chl is a "low" while the sat chl is a "high"

Therefore, at least from a graphic view, there are major discrepancies between satellite and simulation, which rings a sort of alarm bell to me, concerning model performance in general, i.e. also sub-surface.

Next, on line 9 the Authors state that "...remote sensing estimates are generally known to overestimate surface Chl...". So satellite imagery does not necessarily provide a picture of "reality" in sfc Chl, with which the model can be compared.

So where does the truth lie?

Consequently I think that the Authors should comment on this problem more thoroughly, because if satellite imagery does not provide the truth, then why compare the model to the imagery? Or maybe the errors in the imagery are slight with respect to reality, so that model results can be compared to sat chl and e.g. all simulated values that are off by less than XX % from sat values are good? In short, to my opinion the issue is crucial to the rest of the presented results, and thus should not "die out" in two sentences (page 7 lines 9-12) but should be tackled deeply: which simulated values are OK and which aren't?

Line 8. "model surface" -> "the model's surface" By the way, how is the model's surface chlsurf defined? Which model level? Which depth? Line 8 "under-estimates" -> "underestimates" Line 11. "where low Chl values are especially overestimated" -> "where low Chl values especially are overestimated" Line 18. "toward" -> "towards" Line 20. "In this purpose" -> "To this purpose" Line 21. "release" -> "released" Line 22. "figure 4 in the 5 areas defined figure 2" -> "in Figure 4 in the 5 areas defined in Figure 2"

Lines 23. "It shows that the model succeeds in capturing the overall chlorophyll dynamics." I'm afraid not, because: 1) all the relevant non-zero features in the chl profiles are completely different in the model w/respect to the ARGO floats 2) the variability of model results is very low (e.g., the model DCMs vary very little).

In short, I think that the model is inadequate, in that e.g. there are probably some modules which need re-working. My suspects lie in the nutrient "compartment" (is the adoption of the Redfield ratio OK?) and in the treatment of air-sea interaction strength, forcing mixing and consequent nutrient upwelling (are the ECMWF data OK for forcing? This can be easily seen by comparison of model density profiles with Argo density data). That is, if the purely biological compartments of the model are faultless. But I can't be sure.

3.2 Surface chlorophyll phenologies and bioregions

Line 7. "For parsimony... common..." I am not sure what the "parsimony" criterion is. Also, how did you choose the 4 satellite clusters (depicted in Fig.5)? That is, how did you declare a cluster to be "common", given the evident difference in shape among same-colored clusters in Fig. 5? Also, where is the 5th cluster, if Fig. 5 chlsat is practically full?

Line 15. "of cluster to 4" -> "of clusters to 4" Line 15. "Intermittently" -> "Bloom - intermittently" Line 20. "provide" -> "provides" Lines 23-24. "This amplitude is greatest... amplitudes". The chlsat curves in Fig. 5a don't seem to differ much (nor do those in Fig. 5b). Only the blue and yellow curves have slightly higher values in summer than the red and green curves. Line 33. "exit" -> "stand out"

Lines 10-13. Again, I disagree: the inadequacy of the model is also expressed by the important differences between satellite and model clusters, such as the North Ionian being no bloom for the satellite and intermittent bloom for the model.

"3.3 Phytoplankton dynamics in the whole epipelagic layer" -> "3.3. Phytoplankton dynamics in the epipelagic layer"

Line 21. "wide" -> "thick" Line 25. "but an additional maxima now manifests" -> but an additional maximum now appears" Lines 29-30 "in red", "in "blue" -> "red curve", "blue curve"

Line 1. "That result" -> "This result" Line 2. "that Chltot phenology is also different from Chlsurf one" -> " that the Chltot phenology is also different from the Chlsurf one"

3.4 Phytoplankton dynamics in the deep chlorophyll maximum

Line 12. "Gulf of Lion" -> "Gulf of Lions (Fig. 7a)". Another defect of this paper, to my opinion, is that the Authors seldom cite the figure numbers while describing them,

which in some cases can make the reading most cumbersome and difficult. Please correct this where necessary.

Line 13. "at its most oligotrophic" -> "in its most oligotrophic state (Fig. 7b)". Same comment as above.

Line 14. "180 meter" -> "180 meters". BTW why not use "m" in the text? Line 14. "Simulated" -> "The simulated"

Lines 16-18. "Chlorophyll... model". Again, I don't see the realism of the model: there is a very big difference in the model and Argo profiles!

Line 25. "in DCM" -> "in the DCM" Line 27. "The amplitude...". In Fig. 8 the cluster maxima are equal, so I don't understand the statement. Line 27. "receipt" -> "reception"

4 Discussion

4.1 Bioregions and Mixed Layer Depth

Line 4. "ChlSurf" -> "ChlSat", according to definitions Line 5. "Spring-blooms" - "Spring blooms"

4.2 Modelled chlorophyll maximum and total chlorophyll phenologies

Line 30. "figure 2" -> "Figure 2" Line 31. "specific layer" -> "a specific layer". Again, how do you define ARGO Chlsurf? Line 32. "vertical sum of Chl (Chltot; 0 - 300 m, as done in the model)," - > "the 0-300 m Chl integral" Line 33. "(Figure 11)". This figure should maybe come before Figure 10, since it is cited first.

Line 5. "is globally" -> "are globally" Line 10. "what is not seen" -> "which is not seen" Line 16. "being" -> "are" Line 23. "Only differences compare". Unclear Line 23.

"underestimated surface chlorophyll" -> " underestimated model surface chlorophyll"...? Line 24. "As well," -> "Also" Line 26. "when surface" -> "when the surface"

4.3 Underestimated chlorophyll and DCM depth. -> eliminate full stop

4.4 Phytoplankton dynamics in the oligotrophic bioregion.

Line 17. "upward nutrients flux" -> "upward nutrient flux" Line 24. (as DCM" -> "(as the DCM" Lines 27. "phytoplankton biomass" -> "0-300 m phytoplankton biomass integral"

4.5 Surface versus total chlorophyll bioregionalization

Line 8. "that results" -> "that result" Line 21. "intermittently bioregion (yellow)" -> "Bloom-intermittently bioregion (yellow)". To which Figure does this refer? Again cite figures more often, otherwise it's very difficult to follow.

5 Conclusions

Line 18. "coherent, with patterns" Eliminate comma. Line 30-31. "phytoplankton "migration" from the surface layer to the DCM." Since the model provides both Chl and phytoplankton, is this migration seen in the model water column? Is there any evidence of this in situ data studies? Lines 33. "nutrients concentration" -> "nutrient concentration "

Figures and captions

Figure 1 caption. "same 8 years period" -> "same 8-year period"

Figure 3 caption. "model (blue) and satellite estimated" -> "model- (blue) and satellite-estimated"

Figure 3 caption. "The upper pictures (A) include the whole Mediterranean Sea, in the middle (B) the Western basin, and down (C) the Eastern basin (Levantine and Ionian sub-basin)." -> "The (A) entire Mediterranean Sea; (B) Western basin; (C) Eastern basin (Levantine and Ionian sub-basin)." No need for "upper", "middle", etc. since you provide letters to the panels.

Figure 5 caption. "the different clusters resulting" -> "the 4 clusters discussed in the text, resulting" Figure 5 caption. "up", "down" -> "top", "bottom" Please replace in all interested figures ("middle" is OK).

Figure 5, top panels. "chl surf - sat" -> "chlsat" and "chlsurf - PISCES" -> "chlsurf", to be consistent with your definitions in the M&M section.

Figure 6 caption. "the sum of chlorophyll on the 0 - 300 m depth layer" -> "chlorophyll integrated over the 0 - 300 m layer"

Figure 7 caption. "(in ug l-1 ; 10 first meters average)(up)" -> "(in ug l-1); 10 first meters average (top)", "down" -> "bottom"; "in meter ;" -> "in meters;"; "on the water" -> "in the water"

Fig. 8 caption. "maximum chlorophyll layer" -> "maximum chlorophyll depth" (If I understand well).

---

## Author Comment (AC2) · 22 Jan 2019

Now that we received all referees comment, and took a step back from this study for a bit, I can appreciate that (unfortunately for us, the authors) the referee #1 is probably right. It is always hard to get a critical review after a lot a effort, and we – I – put a lot of effort in this study, and it seems that it is not yet over. The authors would like to apology for our vigorous first answer, and would like to thanks the referee #1 for his/her effort in reviewing this paper. I still think that our model could be used in this study, at least as a first attempt. But if 3 referees have found that our model is not convincing, we have go on and find another – more realistic – model for this study. It will not be an easy task,

but we are already on this new quest to find the right (at least better) Mediterranean biogeochemical model. Hoping you will find it more adequate, would you review the next version of this paper.

---

## Author Comment (AC3) · 22 Jan 2019

The authors would like first to thank the Referee #2 for his/her effort in reviewing this study. The paper will very probably be rejected, but I would like to reply anyway to some of your comments. Working back on BGC-ARGO, to prepare the corrected version of the paper, I realised that the chlorophyll field needed corrections that were not included in the data-set I first used. The correction decreases the BGC-ARGO chlorophyll fields by a factor 2 as explained in Barbieux et al. (2018). This reduces the difference between observed and modelled chlorophyll concentration. The model Chl bias is then less important than said in the paper, but this has obviously no impact on

the phenologies, on the too deep modelled DCM, or on the model-satellite differences. About the satellite data, we know there are newer data-set. We tried to do the analysis with Volpe et al. (2007). Mediterranean satellite product, but for an obscure reason we got weird artefacts on the clusters derived from it. As I could not get rid of these artefacts, I finally worked with the Bosc et al. (2004) Mediterranean data-set. Although using a Mediterranean satellite data-set enable to improve the Chl concentration with a better estimate of the CDOM, it "seems" to not have a big impact on the phenology (for instance D'Ortenzio and Ribera d'Alcala (2009) did their Mediterranean bioregion analysis using SeaWIFS – But the CDOM impact on phenology has not been shown as far as I know). About the model, we have to hear the critics, and accept the fact that it is not considered realistic enough for this study. We then have no other alternative than to find a better biogeochemical model and do the analysis again. But finding a more realistic model might not be obvious.

– References:

Barbieux, M., Uitz, J., Gentili, B., Pasqueron de Fommervault, O., Mignot, A., Poteau, A., Schmechtig, C., Taillandier, V., Leymarie, E., Penkerc'h, C., D'Ortenzio, F., Claustre, H., and Bricaud, A.: Bio-optical characterization of subsurface chlorophyll maxima in the Mediterranean Sea from a Biogeochemical-Argo float database, Biogeosciences Discuss., https://doi.org/10.5194/bg-2018-367, in review, 2018.

Bosc, E., Bricaud, A., and Antoine, D.: Seasonal and interannual variability in algal biomass and primary production in the Mediterranean Sea, as derived from 4 years of SeaWiFS observations, Global Biogeochemical Cycles, 18, GB1005, 2004.

D'Ortenzio, F. and Ribera d'Alcalà, M.: On the trophic regimes of the Mediterranean Sea: a satellite analysis, Biogeosciences, 6, 139–148, 2009.

Volpe, G., Santoleri, R., Vellucci, V., Ribera d'Alcala, M., Marullo, S., & D'Ortenzio, F. The colour of the Mediterranean Sea: Global versus regional bio-optical algorithms evaluation and implication for satellite chlorophyll estimates. Remote Sensing of Environment, 107, 625–638, 2007.

---

## Author Comment (AC4) · 22 Jan 2019

I would like to thank the referee #3 for his/her effort in reviewing this study, for the suggested text suggestions and comments. These are very helpful and will be all included in the next submission of the corrected/improved version of the study.

Although the paper will be rejected, I would like to answer some of your comments.

– The analysis is more qualitative because the clustering process used for the bioregionalization, only looks at the variation of chlorophyll. So phenology is what matter for our study, so I didn't want to add any extra plots: 1- because there are already a lot of

plots in this paper, and 2- because (i think) we provide all needed values to appreciate how close/far is the model compare to both satellite and BGC-ARGO floats.

– Why compare model chlorophyll to satellite estimates ? There are several reasons, of which 1- it is not that bad. Of course we have to keep in mind they are estimates, there are errors, but these errors remain low compare to BGC model chlorophyll. Chlorophyll is probably the least well modelled variable (Kwiatkowski et al., 2014). 2- Satellite estimates are basin wide, and high frequency (every 8 days), what is extremely useful for model comparison and diagnostics.

– model surface Chl is the first level concentration.

– "My suspects lie in the nutrient "compartment" (is the adoption of the Redfield ratio OK?)" - that's what I think as well. The circulation model is to be blamed for part of the problem (see the appendix), and I think Redfield ratio is to be blamed in the organic matter remineralization, not necesseraly in phytoplankton production (see my PhD thesis – if you read French – Palmieri (2014)).

– " the inadequacy of the model is also expressed by the important differences between satellite and model clusters, such as the North Ionian" - About the North Ionian there is a full paragraph in the discussion to explain this difference in the modelled Bloom cluster. That's the part where the circulation model is to be blamed.

Hope the revised/corrected version will not be too long to do. Next step is to find a Mediterranean BGC model with a more realistic chlorophyll field.

– References:

Kwiatkowski, L., Yool, A., Allen, J. I., Anderson, T. R., Barciela, R., Buitenhuis, E. T., Butenschön, M., Enright, C., Halloran, P. R., Le Quéré, C., de Mora, L., Racault, M.-F., Sinha, B., Totterdell, I. J., and Cox, P. M.: iMarNet: an ocean biogeochemistry model intercomparison project within a common physical ocean modelling framework, Biogeosciences, 11, 7291-7304, https://doi.org/10.5194/bg-11-7291-2014, 2014.

[Figure]

Palmiéri, J.: Biogeochemical modelling of the Mediterranean Sea, with the NEMO-MED12/PISCES coupled regional model, Ph.D. thesis, Université de Versailles-Saint Quentin en Yvelines, https://tel.archives-ouvertes.fr/tel-01221529, 2014.